# REGULARIZED DEEPIV WITH MODEL SELECTION

## ABSTRACT

In this paper, we study nonparametric estimation of instrumental variable (IV) regressions. While recent advancements in machine learning have introduced flexible methods for IV estimation, they often encounter one or more of the following limitations: (1) restricting the IV regression to be uniquely identified; (2) requiring minimax computation oracle, which is highly unstable in practice; (3) absence of model selection procedure. In this paper, we analyze a Tikhonov-regularized variant of the seminal DeepIV method, called Regularized DeepIV (RDIV) regression, that can converge to the least-norm IV solution, and overcome all three limitations. RDIV consists of two stages: first, we learn the conditional distribution of covariates, and by utilizing the learned distribution, we learn the estimator by minimizing a Tikhonov-regularized loss function. We further show that RDIV allows model selection procedures that can achieve the oracle rates in the misspecified regime. When extended to an iterative estimator, we prove that RDIV matches the current state-of-the-art convergence rate. Furthermore, we conducted numerical experiments to justify the efficiency of RDIV empirically. Our results provide the first rigorous guarantees for the empirically well-established DeepIV method, showcasing the importance of regularization which was absent from the original work.

## 1 INTRODUCTION

Instrumental variable (IV) estimation is an important problem in various fields, such as causal inference (Angrist and Imbens, 1995; Newey and Powell, 2003; Deaner, 2018; Cui et al., 2020; Kallus et al., 2021; 2022), missing data problems (Miao et al., 2018; Wang et al., 2014), dynamic discrete choice models Kalouptsidi et al. (2021) and reinforcement learning (Liao et al., 2021; Uehara et al., 2022a;b; Shi et al., 2022; Wang et al., 2021; Yu et al., 2022).

In this paper, we focus on nonparametric IV (NPIV) regression (Newey and Powell, 2003). NPIV concerns three random variables $X \in \mathbb{R}^d$ (covariate), $Y \in \mathbb{R}$ (outcome variable), and $Z \in \mathbb{R}^d$ (instrumental variables). We are interested in finding a solution $h_0$ of the following conditional moment equation (Dikkala et al., 2020b; Chernozhukov et al., 2019):

$$\mathbb{E}[Y - h(X)|Z] = 0.$$

This is equivalently written as $\mathcal{T}f = r_0$ where $\mathcal{T} : L_2(X) \ni f(X) \mapsto \mathbb{E}[f(X)|Z] \in L_2(Z)$ and $r_0(Z) = \mathbb{E}[Y|Z]$ by denoting $L_2(X), L_2(Z)$ to be the $L_2$ space defined on $X$ and $Z$ with respect to the underlying distribution. Both the operator $\mathcal{T}$ and $\mathbb{E}[Y|Z]$ remain unknown. Hence, we aim to solve $\mathcal{T}f = r_0$ by harnessing an identically independent distributed (i.i.d.) dataset $\{X_i, Y_i, Z_i\}_{i \in [n]}$.

There has been a surge in interest in NPIV regressions that try to integrate general function approximation such as deep neural networks beyond classical nonparametric models (Hartford et al., 2017; Singh et al., 2019; Xu et al., 2021; Zhang et al., 2023; Dikkala et al., 2020b; Bennett and Kallus, 2020; Bennett et al., 2023a;b; Kallus et al., 2022; Singh, 2020). Despite these extensive efforts, existing approaches encounter several challenges. The first challenge is the ill-posedness of the inverse problem. Many existing works (Liao et al., 2020a; Newey and Powell, 2003; Florens et al., 2011; Kato et al., 2021) require that the NPIV solution $h_0$ is unique, and further impose quantitative bounds on measures of ill-posedness. However, it is known that the uniqueness assumption is easily violated in practical scenarios, such as weak IV (Andrews and Stock, 2005; Andrews et al., 2019) or proximal causal inference (Kallus et al., 2021). The second challenge involves the reliance on minimax optimization oracles in many methods (Bennett et al., 2023a; Dikkala et al., 2020b; Liao et al.,

Table 1: Summary of IV regression literature with general function approximation such as neural networks. "Model Selection" means allowing model selection methods over any hypothesis space. "No Minimax" means no need of minimax oracle. "No Uniquness" means unique solution is not assumed.

| | Model Selection | No Minimax | No Uniqueness | RMSE rates |
|---|---|---|---|---|
| Chen and Pouzo (2012) | | ✓ | | ✓ |
| Hartford et al. (2017) | | ✓ | | |
| Dikkala et al. (2020a) | | | | |
| Liao et al. (2020a) | | | | ✓ |
| Xu et al. (2021) | | ✓ | | |
| Bennett et al. (2023a) | | | ✓ | ✓ |
| Bennett et al. (2023b) | | | ✓ | ✓ |
| Ours (RDIV) | ✓ | ✓ | ✓ | ✓ |

2020a; Bennett et al., 2023b; Zhang et al., 2023), which results in minimax non-convex non-concave optimization when invoking deep neural networks. However, currently, such an optimization can be notoriously unstable and may fail to converge (Lin et al., 2020b; Jin et al., 2020; Lin et al., 2020a; Diakonikolas et al., 2021; Razaviyayn et al., 2020). Instead, our approach seeks to address this challenge by proposing a computationally efficient estimator that relies on standard supervised learning oracles rather than minimax oracles. The third challenge is the absence of clear procedures for model selection in existing works (Chen and Pouzo, 2012; Xu et al., 2021; Zhang et al., 2023; Cui et al., 2020; Hartford et al., 2017). Such a procedure, including techniques such as cross-validation, has played a pivotal role in modern machine-learning algorithms (Bartlett et al., 2002a; Gold and Sollich, 2003; Guyon et al., 2010; Cawley and Talbot, 2010; Raschka, 2018; Emmert-Streib and Dehmer, 2019; McAllester, 2003). In NPIV problems, model selection becomes essential, particularly in scenarios where the ground-truth solution $h_0$ lies outside the chosen function classes optimized by the algorithm. However, model selection remains an open question for minimax approaches due to having a test function for the inner maximization problem that might change when generalizing from the empirical distribution to population distribution. On the other hand, while some prior works employ a loss minimization approach (e.g. Chen and Pouzo (2012); Zhang et al. (2023)), model selection would be restricted to a specific hypothesis space, such as kernels or sieves, and no theoretical discussion for model selection under general function approximation has been discussed.

In this paper, we propose and analyze a variant of the well-established DeepIV method (Hartford et al., 2017), that addresses the aforementioned challenges, which we refer to as the *Regularized DeepIV (RDIV)*. This approach consists of two steps. First, we learn the operator $\mathcal{T}$ by maximum likelihood estimation (MLE). Secondly, we obtain an estimator for $h_0$ by solving a loss incorporating the learned $\mathcal{T}$ and Tikhonov regularization (Ito and Jin, 2014) to handle scenarios where solutions of the conditional moment constraint are nonunique. While RDIV can be viewed as a regularized variant of the DeepIV method of Hartford et al. (2017) with a non-parametric MLE first-stage, no prior theoretical convergence guarantees exist for the DeepIV method. We show that our estimators can converge to the least norm IV solution (even if solutions are nonunique) and derive its $L_2$ error rate guarantee based on critical radius. Subsequently, we introduce model selection procedures for our estimators. Particularly, we provide theoretical guarantees for model selection via out-of-sample validation approaches, and show an oracle result in our context. Finally, we further illustrate that RDIV can be easily generalized to an iterative estimator that more effectively leverages the well-posedness of $h_0$.

Our contribution is to propose the first formal theoretical results for the well-established NPIV method DeepIV, with an additional Tikhonov regularization. Although simple, such regularization imparts strong convexity to the loss function, thereby enhancing its generalization ability. Specifically, we show that RDIV (a) operates in the absence of the uniqueness assumption, (b) does not rely on the minimax computational oracle, and (c) allows for model selection. Subsequently, we demonstrate that RDIV can be extended to an iterative estimator. We show that our estimators achieve a state-of-the-art convergence rate in terms of $L_2$ error analogous to Bennett et al. (2023b) for the iterative version, as well as the non-iterative version when $h_0$ is well-posed. In contrast, Bennett et al. (2023b) relies on a minimax computational oracle and does not permit us to perform model

selection. Therefore, our estimator can be seen as an estimator with a strong theoretical guarantee due to the property (a) while it is practical due to properties (b) and (c). Notably, none of the existing works can enjoy such a guarantee, as shown in Table 1. From a technical perspective, the key challenge in our proof lies in effectively controlling the density estimation error resulting from the first-stage MLE. This step introduces a density estimation error in Hellinger distance, whereas the final results require bounding the $L_2$ distance between $\hat{h}$ and $h_0$. This task is nontrivial since the estimator is not Neyman orthogonal (Foster and Syrgkanis, 2019), and directly converting the error from the Hellinger distance to $L_2$ norm would lead to a slower convergence rate.

## 2 NOTATIONS

For a function $f : \mathcal{X} \times \mathcal{Y} \times \mathcal{Z} \to \mathbb{R}$, we denote its population expectation by $\mathbb{E}[f(X, Y, Z)]$. We denote the empirical mean of $f$ by $\mathbb{E}_n[f(X, Y, Z)] := \frac{1}{n}\sum_{i=1}^n f(X_i, Y_i, Z_i)$. We denote the set of all probability distributions defined on set $\Omega$ by $\Delta(\Omega)$. We denote the $L_p$ norm of $f$ by $\|f\|_p := \mathbb{E}[|f|^p]^{1/p}$. Throughout the paper, whenever we use a generic norm of a function $\|f\|$, we will be referring to the $L_2$-norm. For two density function $p(x)$ and $q(x)$, we denote their Hellinger distance by $H(p(\cdot) \mid q(\cdot)) = \int_{\mathcal{X}}(\sqrt{p(x)} - \sqrt{q(x)})^2 d\mu(x)$. For a functional operator $\mathcal{T} : L_2(X) \to L_2(Z)$, we denote the range space of $\mathcal{T}$ by $\mathcal{R}(\mathcal{T})$, i.e., $\mathcal{R}(\mathcal{T}) = \{\mathcal{T}h : h \in L_2(X)\}$. Moreover, we use $\mathcal{T}^* : L_2(Z) \to L_2(X)$ to denote the adjoint operator of $\mathcal{T}$, i.e., $\langle g, \mathcal{T}h \rangle_{L_2(Z)} = \langle \mathcal{T}^*g, h \rangle_{L_2(X)}$ for any $h \in L_2(X), g \in L_2(Z)$, where $\langle \cdot, \cdot \rangle_{L_2(X)}$ and $\langle \cdot, \cdot \rangle_{L_2(Z)}$ are inner products over $L_2(X)$ and $L_2(Z)$, respectively. For $\theta \in \Theta = \{\theta | \sum_j \theta_j = 1, \theta_j \geq 0, \forall j\}$, we denote $h_\theta = \sum_j \theta_j h_j$. We use $e_j$ to denote the one-hot vector where that is zero except for the $j^{th}$ component, which equals to 1. For a function class $\mathcal{F}$, we define the localized Rademacher complexity by $\bar{R}_n(\delta; \mathcal{F}) := \mathbb{E}\big[\mathbb{E}_\epsilon\big[\sup_{f \in \mathcal{F}, \|f\|_2 \leq \delta}\big|\frac{1}{n}\sum_{i=1}^n \epsilon_i f(x_i, z_i)\big|\big]\big]$, where $\epsilon_i$ are i.i.d. Rademacher random variables. For a function class $\mathcal{F}$ over $\mathcal{X}$ and $\mathcal{Z}$, we define its star hull by $\mathrm{star}(\mathcal{F}) = \{\gamma f, \gamma \in [0, 1], f \in \mathcal{F}\}$. For a function class $\mathcal{F}$, we denote $\bar{\mathcal{F}} := \mathrm{star}(\mathcal{F} - \mathcal{F})$ to define its symmetrized star hull. We define the critical radius $\delta_{n,\mathcal{F}}$ of a function class $\mathcal{F}$ as any solution to the inequality $\delta^2 \geq \bar{R}_n(\mathrm{star}(\mathcal{F} - \mathcal{F}), \delta)$. We use $\mu$ to denote the Lebesgue measure.

## 3 PROBLEM STATEMENT AND PRELIMINARIES

As mentioned in Section 1, we aim to solve the following inverse problem with respect to $h$, known as the nonparametric IV regression:

$$\mathcal{T}h = r_0, \quad r_0 := \mathbb{E}[Y|Z]. \tag{1}$$

While $\mathcal{T}$ and $r_0$ are unknown a priori, using i.i.d. observations $\{X_i, Y_i, Z_i\}_{i \in [n]}$, we aim to solve this equation. We denote its associated distributions by $g_0$, e.g., denote the conditional density of $X \in \mathcal{X}$ given $Z \in \mathcal{Z}$ by $g_0(x|z) \in \{\mathcal{X} \times \mathcal{Z} \to \mathbb{R}\}$. Throughout this work, we assume a solution to Equation equation 1 exists.

**Assumption 1** (Existence of Solutions). *We have $r_0 \in \mathcal{R}(\mathcal{T})$, i.e. $\mathcal{N}_{r_0}(\mathcal{T}) := \{h \in \mathcal{H} : \mathcal{T}h = r_0\} \neq \varnothing$.*

Crucially, even though a solution to equation 1 exists, it might not be unique. Hence, we propose to target a specific solution that achieves the least norm, defined as:

$$h_0 := \arg\min_{h \in \mathcal{N}_{r_0}(\mathcal{T})} \|h\|_2. \tag{2}$$

Note this least norm solution is well-defined, as it is defined by the projection of the origin onto a closed affine space $\mathcal{N}_{r_0}(\mathcal{T}) \subset L_2(X)$. Indeed, with Assumption 1, it is easy to prove that $h_0$ in equation 2 always exists (Bennett et al., 2023a, Lemma 1).

As we emphasize the challenges in Section 1, although there have been a lot of method that use minimax optimization for estimating $h_0$, when using general function approximation such as neural networks, the minimax optimization tends to be computationally hard (Lin et al., 2020b; Jin et al., 2020; Lin et al., 2020a; Diakonikolas et al., 2021; Razaviyayn et al., 2020). Moreover, it remains unclear how to perform model selection for those methods. Hence, in this paper, we aim to propose a new method that can incorporate any function approximation for estimating the least square norm solution $h_0$ in equation 2 with a strong convergence guarantee in $L_2(X)$ under mild assumptions (i.e., such as without the uniqueness of $h_0$) while allowing for model selection.

---

**Algorithm 1** Regularized Deep IV (RDIV)

---

**Require:** Validation dataset $\{X_i, Y_i, Z_i\}_{i \in [n']}$ that is independent from the training dataset, function class $\mathcal{G} \subset \{\mathcal{Z} \to \Delta(\mathcal{X})\}$, function class $\mathcal{H} \subset \{\mathcal{X} \to \mathbb{R}\}$, a regularization hyperparameter $\alpha \in \mathbb{R}_{>0}$

1: Learn $\hat{g}(x|z)$ with MLE:

$$\hat{g} = \arg\max_{g \in \mathcal{G}} \mathbb{E}_n[\log g(X|Z)], \tag{4}$$

2: Learn $\hat{h}$ by the following estimator:

$$\hat{h} = \arg\min_{h \in \mathcal{H}} \mathbb{E}_n[(Y - (\hat{\mathcal{T}}h)(Z))^2] + \alpha \cdot \mathbb{E}_n[h(X)^2] \tag{5}$$

where $\hat{\mathcal{T}} : L_2(X) \to L_2(Z)$ is defined by $\hat{\mathcal{T}}f(Z) = \mathbb{E}_{x \sim \hat{g}(X|Z)}[f(X)]$ using $\hat{g}$ in the first step.

**output** $\hat{h}$.

---

## 4 REGULARIZED DEEP IV

In this section, we introduce a two-stage algorithm, Regularized DeepIV (RDIV), aimed at obtaining the least square solution $h_0$ as defined in Equation equation 2. Even though we borrow the DeepIV terminology from the prior work (Hartford et al., 2017), our method can be used with arbitrary function approximators and not necessarily neural network function spaces. Being inspired by the original constrained optimization equation 2, we aim to solve a regularized version of the problem, shown by the following:

$$h_* := \arg\min_{h \in \mathcal{H}} \|Y - \mathcal{T}h\|_2^2 + \alpha\|h\|_2^2 \tag{3}$$

where $\mathcal{H} \subset L_2(X)$ represents a hypothesis class that consists of possible candidates for $h_0$, and $\alpha \in \mathbb{R}^+$ denotes a parameter controlling the strength of regularization. While this formulation itself has been known in the literature on general inverse problems (Cavalier, 2011; Mendelson and Neeman, 2010), we consider common scenarios in IV where both the conditional expectation operator $\mathcal{T}$ and the population expectation in Equation equation 3 are unknown, and need to leverage dataset $\{X_i, Y_i, Z_i\}$.

To address this challenge, by integrating general function approximation such as neural networks, we introduce a two-stage method, the *Regularized Deep Instrumental Variable (RDIV)*, which is summarized in Algorithm 1. In the first stage, given a function class $\mathcal{G}$ comprising functions of the form $\{g : \mathcal{X} \times \mathcal{Z} \to \mathbb{R}, \int_{\mathcal{X}} g(x|z)\mu(dx) = 1 \text{ for all } z\}$, we aim to learn the conditional expectation operator $\mathcal{T}$ by estimating the ground-truth conditional density $g_0(x|z)$ from the dataset $\{X_i, Z_i\}_{i \in [n]}$ with MLE in Equation equation 4. In the second stage, with the learned conditional density $\hat{g}$ in the first step, we learn $h_0$ by replacing expectation and $\mathcal{T}$ in Equation equation 3 with empirical approximation and $\hat{\mathcal{T}}$, respectively, as shown in Equation equation 5.

Importantly, similar to DeepIV, RDIV does not necessitate a demanding computational oracle such as non-convex non-concave minimax or bilevel optimization, unlike many existing works for non-parametric IV with general function approximation (Lewis and Syrgkanis, 2018; Xu et al., 2021; Bennett et al., 2023a). Even when using neural networks for $\mathcal{G}$ and $\mathcal{H}$, we just need standard ERM oracles for density estimation or regression whose optimization is empirically known to be successful and theoretically more supported (Du, 2019; Chen et al., 2018; Zaheer et al., 2018; Barakat and Bianchi, 2021; Wu et al., 2019; Zhou et al., 2018; Ward et al., 2020). We leave the numerical comparison between our method and existing NPIV methods (Hartford et al., 2017; Dikkala et al., 2020b; Xu et al., 2021; Singh et al., 2019) in Appendix 9.

**Remark 1** (Comparison with DeepIV (Hartford et al., 2017))**.** *A key distinction between RDIV and the original DeepIV (Hartford et al., 2017) lies in our introduction of an explicit regularization term in Equation equation 5. Such a term endows the loss function with strong convexity, which plays a pivotal role in obtaining guarantees without the requirement for solution uniqueness. Furthermore, Hartford et al. (2017) lacks a rigorous discussion on convergence guarantees or model selection. Our contributions primarily focus on the theoretical aspect, showcasing rapid convergence guaran-*

*tees under mild assumptions, linking them to a formal model selection procedure, and exploring the iterative version to achieve a refined rate in Section 8.*

**Remark 2** (Computaion for $\hat{\mathcal{T}}$). *Some astute readers might notice it could be hard to evaluate $\hat{\mathcal{T}}h$ exactly in Equation equation 5. However, in practical application when $h$ is parametrized as a neural network, we can sample a batch of $\{X'_j\}_{j \in [B]}$ by $\hat{g}(X|Z_i)$ for every $Z_i$ in the dataset, and calculate a stochastic gradient that is an unbiased estimator of the real gradient of the loss function in Equation equation 5. Existing theory and empirical results for stochastic first-order methods can then guarantee the performance in many scenarios (Jin et al., 2019; Barakat and Bianchi, 2021; Chen et al., 2018; Hartford et al., 2017).*

## 5 FINITE SAMPLE GUARANTEES

In this section, we demonstrate a convergence result of our estimator $\hat{h}$ in RDIV to $h_0$ and derive its $L_2$ error rate after introducing several assumptions.

We commence by introducing the $\beta$-source condition, a concept commonly used in the literature on inverse problems (Carrasco et al., 2007; Ito and Jin, 2014; Engl et al., 1996; Bennett et al., 2023b; Liao et al., 2021), which mathematically captures the well-posedness of the function $h_0$.

**Assumption 2** ($\beta$-Source Conditon). *The least norm solution $h_0$ satisfies $h_0 = (\mathcal{T}^*\mathcal{T})^{\beta/2}w_0$ for some $w_0 \in \mathcal{H}$ and $\beta \in \mathbb{R}_{\geq 0}$, i.e., $h_0 \in \mathcal{R}(\mathcal{T}^*\mathcal{T})^{\beta/2}$. Recall $\mathcal{T}^*$ is an adjoint operator of $\mathcal{T}$ defined in Section 2.*

In the following, we present its interpretation. First, as special cases, when $\mathcal{X}, \mathcal{Z}$ are finite (e.g., discrete random variables), it holds when $\beta = \infty$. However, in our cases of interests where $\mathcal{X}, \mathcal{Z}$ are not finite, this assumption restricts the smoothness of $h_0$. Intuitively, when the parameter $\beta$ is large, the function $h_0$ exhibits greater smoothness, and the assumption gets stronger, in the sense that eigenfunctions of $h_0$ relative to an operator $\mathcal{T}$ have smaller eigenvalues as explained in Bennett et al. (2023a, Section 6.4).

Next, we introduce another standard assumption as follows. This requires that the function classes $\mathcal{H}$ and $\mathcal{G}$ are well-specified. We will later consider misspecified cases as in Section 6.

**Assumption 3** (Realizability of function classes). *We assume $h_0 \in \mathcal{H}$, $g_0 \in \mathcal{G}$.*

The final assumption is as follows. This is standard in analyzing the convergence of nonparametric MLE (Wainwright, 2019, Chap 14, p.g. 476). We will later discuss how to relax such an assumption Appendix C.

**Assumption 4** (Lower-bounded density). *We assume a constant $C_0 > 0$ such that $g_0(x|z) > C_0$ holds for all $x \in \mathcal{X}$ and $z \in \mathcal{Z}$.*

Finally, we present our guarantee for Algorithm 1.

**Theorem 5** ($L_2$ convergence rate for RDIV with MLE). *Suppose Assumption 2,3,4 hold. Let $\|Y\|_\infty \leq C_Y$, $\|h\|_\infty \leq C_\mathcal{H}$ holds for all $h \in \mathcal{H}$, $\|g\|_\infty \leq C_\mathcal{G}$ holds for all $g \in \mathcal{G}$. There exists absolute constant $c_1, c_2$, such that with probability at least $1 - c_1 \exp(c_2 n \delta_n^2)$:*

$$\|\hat{h} - h_0\|_2^2 = O(\underbrace{\delta_n^2/\alpha^2}_{(i)} + \underbrace{\alpha^{\min(\beta,2)}}_{(ii)})$$

*In particular, by setting $\alpha = \delta_n^{\frac{2}{2+\min\{\beta,2\}}}$ we have*

$$\|\hat{h} - h_0\|_2^2 = O\big(\delta_n^{\frac{2\min\{\beta,2\}}{2+\min\{\beta,2\}}}\big). \tag{6}$$

*Here $\delta_n = \max\{\delta_{n,\mathcal{G}}, \delta_{n,\mathcal{H}}\}$, where $\delta_{n,\mathcal{F}}$ is the critical radius of $star(\mathcal{F} - \mathcal{F}) = \{\lambda(f - f'), f, f' \in \mathcal{F}, \lambda \in [0,1]\}$. $O(\cdot)$ hides constants of polynomial order of $C_Y, C_\mathcal{G}, C_\mathcal{H}$, and $1/C_0$.*

The critical radius $\delta_n$ measures the statistical complexity of function class $\mathcal{H}$ and $\mathcal{G}$. For example, for parametric class or Gaussian Kernel, $\delta_n = \tilde{O}(n^{-1/2})$, while for first order Sobolev class, $\delta_n = \tilde{O}(n^{-1/3})$ (Wainwright, 2019; Bartlett et al., 2002b). In those cases, when $\beta \geq 2$, the final rate in $L_2$ metric will be $\tilde{O}(n^{-1/2})$ in the former case and $\tilde{O}(n^{-1/3})$ in the latter case, respectively. Note that when the complexity of the function class is known, the regularization constant $\alpha$ can be directly calculated by Theorem 5. We now give the interpretation of our result. The bound of $\|\hat{h} - h_0\|_2^2$ consists of two terms. Term (i) comes from a statistical error to estimate $h_*$ from $\mathcal{H}$ and $\mathcal{G}$ (i.e.,

$\|\hat{h} - h_*\|_2$). Here, we use the strong convexity owing to Tikhonov regularization as it enables us to convert the population risk error to an error in $L_2$ metric. Then, we properly bounded the population risk from above by the empirical process term properly. While this $\delta_n^2$ rate is known as the standard fast rate in nonparametric regression (Wainwright, 2019), our result is still non-trivial because we need to handle a statistical error term properly when approximating $\mathcal{T}$ with $\hat{\mathcal{T}}$, which comes from the MLE error in the form of Hellinger distance.

The term (ii) comes from the bias $\|h_0 - h_*\|_2$ incurred by adding a Tikhonov regularization. This analysis has been used in existing works (e.g., (Cavalier, 2011)). Due to $\min(\beta, 2)$, while we cannot leverage a high smoothness $\beta$ especially when $\beta \geq 2$, we will see how to leverage $\beta$ in such a case by introducing an iterative estimator in Section 8.

We also compare our work to existing state-of-the-art convergence rate $O\big(\delta_n^{2\frac{\min\{\beta,1\}}{1+\min\{\beta,1\}}}\big)$ in Bennett et al. (2023b), in which they employ a minimax-type algorithm. When $\beta \geq 2$, i.e., $h_0$ is well-posed, we achieve the same rate. We also remark that although our rate is slightly slower than theirs when $\beta \leq 2$, our method does not require a minimax-optimization oracle and can be incorporated with method selection methods. Besides, we will show that our method can achieve a state-of-the-art rate in our extension to iterative estimator in Section 8.

# 6 MISSPECIFIED SETTING

Next, we establish the finite sample result when Assumption 3 does not hold, i.e., function classes $\mathcal{H}$ and $\mathcal{G}$ are misspecified. This result serves as an important role in formalizing the model selection procedure in Section 7.

**Theorem 6** ($L_2$ convergence rate for RDIV with MLE under misspecification). *Suppose Assumption 2 and 4 hold, and there exists $h^\dagger \in \mathcal{H}$ and $g^\dagger \in \mathcal{G}$ such that $\|h_0 - h^\dagger\|_2 \leq \epsilon_{\mathcal{H}}$ and $\mathbb{E}_{z \sim g_0}[D_{KL}(g_0(\cdot|z) \mid g^\dagger(\cdot|z))] \leq \epsilon_{\mathcal{G}}$. For any $0 < \alpha \leq 1$, we have*

$$\|\hat{h} - h_0\|_2^2 = O\bigg( \underbrace{\frac{\delta_n^2}{\alpha^2}}_{(b1)} + \underbrace{\alpha^{\min\{\beta+1,2\}-1}}_{(b2)} + \underbrace{\frac{\epsilon_{\mathcal{H}}^2}{\alpha} + \frac{\epsilon_{\mathcal{G}}}{\alpha^2}}_{(b3)} \bigg)$$

*holds with probability at least $1 - c_1 \exp(c_2 n \delta_n^2)$. Here $\delta_n$ has the same definition in Theorem 5.*

The bound for $\|\hat{h} - h_0\|_2^2$ consists of three terms: term (b1) measures the statistical deviation of a normalized empirical process, term (b2) measures the regularization error caused by Tikhonov regularization and term (b3) measures the effect of model misspecification. Here term (b3) has a $\text{poly}(\frac{1}{\alpha})$ dependency. This is because model misspecification causes a higher population risk in both stage 1 and 2 of Algorithm 1. Hence, the more convex the loss function, the lesser the shift in the optimizer. The readers may notice that term (b2) is slightly slower than the original bias term in Theorem 8. This is because the difference of the optimal value in equation 3 due to misspecification of $\mathcal{H}$ is of order $O(\alpha^{\min\{\beta+1,2\}} + \epsilon_{\mathcal{H}}^2)$, as we will show in Lemma 2 in the Appendix. By the $\alpha$-strong convexity endowed by Tikhonov regularization, this results in a shift of $h_*$ of magnitude $O\big(\alpha^{\min\{\beta+1,2\}-1} + \epsilon_{\mathcal{H}}^2/\alpha\big)$.

Theorem 6 is particularly useful when we apply estimators based on sample-dependent function classes $\mathcal{H}$ and $\mathcal{G}$ (e.g. sieve estimators) that approximate certain function spaces. For example, $\mathcal{H}$ can be linear models with polynomial basis functions that take the form $\langle \phi(X), \theta \rangle$, which can gradually approach Hölder or Sobolev balls, and $\mathcal{G}$ can be a set of neural networks with a growing dimension (Chen, 2007; Chen et al., 2022; Schmidt-Hieber, 2020). More specifically, when $X$ and $Z$ are bounded, and $h_0$ and $g_0$ are $s$-Hölder smooth, it is well known that a deep ReLU neural network with depth $O(\log(1/\epsilon))$, width $O(d\epsilon^{-d/s})$ and weights bounded by $\tilde{O}(1)$ could satisfy the approximation error in Theorem 6 (Schmidt-Hieber, 2019), recall that $d$ is the dimension of $X$ and $Z$. In that case, $\delta_n^2 = \tilde{O}(\epsilon^{-d/s}/n)$ (Bartlett et al., 2002b; Chen et al., 2022). Choosing the architecture of the neural network according to $\epsilon = \tilde{O}(n^{-1/(1+d/s)})$, then Theorem 6 shows that by setting $\alpha = O(n^{\frac{1}{(1+d/\alpha)(\min\{\beta+1,2\}+1)}})$, we have $\|\hat{h} - h_0\|_2^2 = \tilde{O}(n^{\frac{\min\{\beta+1,2\}-1}{(1+d/s)(\min\{\beta+1,2\}+1)}})$.

---

**Algorithm 2** Model Selection for Regularized Deep IV

---

**Require:** Validation dataset $\{X_i', Y_i', Z_i'\}_{i \in [n]}$, $M$ candidate models $\{h_i\}_{i=1}^M$, a regularization hyperparameter $\alpha \in \mathbb{R}_{>0}$, an estimator $\hat{g}$, which can obtained by MLE with standard model selection procedure in Birgé (2006); Cohen and Pennec (2011).

1: Learn $\hat{\theta}$ with each of the followings:

$$\textbf{Best-ERM:} \quad \hat{\theta} = \underset{\theta = e_1, \ldots, e_M}{\arg\min} \; \mathbb{E}_n[(Y - (\hat{\mathcal{T}} h_\theta)(Z))^2] + \alpha \cdot \mathbb{E}_n[h_\theta(X)^2], \quad (7)$$

$$\textbf{Convex-ERM:} \quad \hat{\theta} = \underset{\theta \in \Theta}{\arg\min} \; \mathbb{E}_n[(Y - (\hat{\mathcal{T}} h_\theta)(Z))^2] + \alpha \cdot \mathbb{E}_n[h_\theta(X)^2], \quad (8)$$

where $h_\theta = \sum_{j=1}^M \theta_i h_i$, $\sum_{j=1}^M \theta_j = 1, \theta_j \geq 0$, $\hat{\mathcal{T}} f(Z) = \mathbb{E}_{x \sim \hat{g}(X|Z)}[f(X)]$ and $\mathbb{E}_n[\cdot]$ is defined for $\{X_i', Y_i', Z_i'\}_{i \in [n]}$.

**output** $h_{\hat{\theta}}$.

---

## 7 MODEL SELECTION

One advantage of employing the proposed two-staged algorithm is that it enables model selection, which is not attainable when a minimax approach is used. In this section, we explain how we perform model selection. We focus on the model selection for the second stage, as the conditional density $\hat{g}$ from the first stage can be selected via existing methods for model selection for maximum likelihood estimators (e.g. Birgé (2006); Cohen and Pennec (2011); Vijaykumar (2021)).

With an MLE-based estimator $\hat{g}$ obtained from the first stage in Algorithm 2, we consider model selection using the regularized loss in the second stage, with theoretical guarantees in the $\|\cdot\|_2$ metric. More concretely, given a choice of $M$ candidate models $\{h_1, \ldots, h_M\}$ and a validation dataset $\{X_i', Y_i', Z_i'\}_{i=1}^n$ (distinct from the one used for training models $\{h_i\}$ and $\hat{g}$), the goal is for the final output of the model selection algorithm to achieve oracle rates with respect to the minimal misspecification error.

We present our algorithm in Algorithm 2. We provide two options for model selection: Best-ERM and Convex-ERM. Best-ERM selects the model that minimizes the regularized loss on a validation set, while Convex-ERM constructs a convex aggregate of the candidate models that minimizes the regularized loss on a validation set.

**Theorem 7** (Model Selection Rates). *Consider the model selection problem given $M$ candidate models with any choice of $\alpha$, over $M$ function classes $\{\mathcal{H}_1, \ldots, \mathcal{H}_M\}$. Suppose Assumption 2 and 4 hold, and there exists $g^\dagger \in \mathcal{G}$ and $h_j^\dagger \in \mathcal{H}_j$ for all $j$ such that $\|h_0 - h_j^\dagger\|_2 \leq \epsilon_{\mathcal{H}_j}$ and $\mathbb{E}\left[\int_{\mathcal{X}}(g^\dagger(x|Z) - g_0(x|Z))^2 d\mu(x)\right] \leq \epsilon_{\mathcal{G}}$. Assume that $Y$ is almost surely bounded by $C_Y$, each candidate model $h_j$ is uniformly bounded in $[-C_{\mathcal{H}}, C_{\mathcal{H}}]$ almost surely. Let $\delta_{n,j} = \max\{\delta_{n,\mathcal{G}}, \delta_{n,\mathcal{H}_j}, \delta_{n,M}\}$, where $\delta_{n,M}$ denotes the critical radius of the convex hull over $M$ variables for Best-ERM (i.e. $\delta_{n,M} = \frac{\log(M)}{n}$), and the critical radius of the set of $M$ candidate functions for Convex-ERM (i.e. $\delta_{n,M} = \frac{M}{n}$).*

*With probability $1 - c_1 \exp(c_2 n \sum_j^M \delta_{n,j}^2)$, the output of Convex-ERM or Best-ERM $\hat{\theta}$, satisfies:*

$$\|h_{\hat{\theta}} - h_0\|_2^2 \leq \min_{j \in [M]} O\left(\frac{\delta_{n,j}^2}{\alpha^2} + \alpha^{\min\{\beta+1,2\}-1} + \frac{\epsilon_{\mathcal{H}_j}^2}{\alpha} + \frac{\epsilon_{\mathcal{G}}}{\alpha^2}.\right)$$

We explain its implications. Most importantly, our obtained rate is the best (i.e., oracle rate) among rates when invoking a result of (convergence result for RDIV in Theorem 6 with misspecified model) for each function class $\mathcal{H}_i$. Some astute readers might wonder whether we can just invoke Theorem 6 by making new function classes $\mathcal{H}_{\text{best}} := \{h_\theta : \theta = e_1, \ldots, e_M\}$ or $\mathcal{H}_{\text{conv}} := \{h_\theta : \sum_j \theta_j = 1, \theta_j \geq 0\}$, and bound the misspecification error $\epsilon_{\mathcal{H}_{\text{conv}}}$ or $\epsilon_{\mathcal{H}_{\text{best}}}$ by $\|h_j - h_0\|$ will lead to a slower rate with an extra factor of $\frac{1}{\alpha}$. The key is only to handle the misspecification error once to avoid the $\frac{1}{\alpha}$ factor by deferring the invocation of strong convexity and working with the excess risk (difference in the expected loss) instead of the $L_2$ difference.

---

**Algorithm 3** Iterative Regularized Deep IV

---

**Require:** Dataset $\{X_i, Y_i, Z_i\}_{i \in [n]}$, function class $\mathcal{G}$, function class $\mathcal{H}$, $\hat{h}_{-1} = 0$
1: Learn $\hat{g}(x|z)$ by MLE equation 4
2: **for** $m = 1, 2, \cdots, M$ **do**
3:   Learn $\hat{h}_m$ by iterative Tikhonov estimator as the following:

$$\hat{h}_m = \underset{h \in \mathcal{H}}{\arg\min} \, \mathbb{E}_n[(Y - \hat{\mathcal{T}}h(Z))^2] + \alpha \cdot \mathbb{E}_n[(h(X) - \hat{h}_{m-1}(X))^2], \qquad (10)$$

4: **end for**
**output** $\hat{h}_M$

---

# 8 EXTENSION TO ITERATIVE VERSION

One drawback of the result so far is its lack of adaptability to the degree of ill-posedness in the inverse problem, especially for larger values of $\beta$ corresponding to milder problems, when $\beta \geq 2$. To address this issue, in this section, we further generalize our results in Section 4 and 5, and propose an iterated Regularized Deep method, which is summarized in Algorithm 3. In this algorithm, instead of targeting equation 3, we target $h_{m,*}$, which is given by the following recursive least square regression with Tikhonov regularization:

$$h_{m,*} = \underset{h \in \mathcal{H}}{\arg\min} \, \mathbb{E}[(Y - \mathcal{T}h(Z))^2] + \alpha \cdot \mathbb{E}[(h - h_{m-1,*})^2(X)]. \qquad (9)$$

and we set $h_{-1,*} = 0$. This is the recursive version of the previous regularized objective in Equation equation 3, by using Tikhonov regularization around a prior target $h_{m-1,*}$ instead of 0. Then, with the learned conditional density $\hat{g}$ by MLE in Equation equation 4, we construct an estimator in equation 10 by replacing expectation and an operator $\mathcal{T}$ with empirical approximation and the learned operator $\hat{\mathcal{T}}$, respectively, in Equation equation 9.

Now, we delve into estimating the finite sample convergence rate of Algorithm 3. Our findings are summarized in the following theorem.

**Theorem 8** ($L_2$ convergence rate for iterative MLE estimator). *Suppose Assumption 2, 3, 4 hold. Let $\|Y\|_\infty \leq C_Y$, $\|h\|_\infty \leq C_\mathcal{H}$ holds for all $h \in \mathcal{H}$, $\|g\|_\infty \leq C_\mathcal{G}$ holds for all $g \in \mathcal{G}$. By setting $\alpha = \delta_n^{\frac{2}{2+\min\{\beta, 2m\}}}$, with probability at least $1 - c_1 m \exp(c_2 n \delta_n^2)$, we have*

$$\|\hat{h}_m - h_0\|_2^2 = O\big(16^{2m} \cdot \delta_n^{\frac{2\min\{\beta, 2m\}}{2+\min\{\beta, 2m\}}}\big),$$

*here $\delta_n$ has the same definition in Theorem 5.*

Importantly, we can have a rate $O\big(\delta_n^{\frac{2\beta}{2+\beta}}\big)$ in relatively mild conditions while the previous Theorem 5 (non-iteratie version) can only allow for $O\big(\delta_n^{\frac{2\min(\beta,2)}{2+2\min(\beta,2)}}\big)$, and cannot fully leverage the well-posedness of $h_0$, illustrated by the source condtion $\beta$. Indeed, if we choose the iteration number $m = \lceil \min\{\beta/2, \log\log(1/\delta_n)\} \rceil$, then we get a rate of

$$\|\hat{h}_m - h_0\|_2^2 = O\bigg(\min\{16^\beta, \log(1/\delta_n)\} \delta_n^{\frac{2\min\{\beta, 2m\}}{2+\min\{\beta, 2m\}}}\bigg).$$

Hence for any constant $\beta$, as $n$ grows, eventually $\log\log 1/\delta_n \geq \beta$, and we get the rate of $O\big(\delta_n^{\frac{2\beta}{2+\beta}}\big)$. This rate can be achieved even if $\beta$ grows with $n$, as long as it grows slower than $O(\log\log 1/\delta_n)$. If $\delta_n = O(n^{-\iota})$ for some $\iota > 0$, e.g. RKHS or first order Sobolev space (Wainwright, 2019, Chapt 14.1.2), then we note that we can set $m = \lceil \min\{\beta/2, \sqrt{\log(1/\delta_n)}\} \rceil$, and $16^{\sqrt{\log(1/\delta_n)}} = O(n^\epsilon)$ for any $\epsilon > 0$, thus we still obtain a rate of $O\big(\delta_n^{\frac{2\beta}{2+\beta}}\big)$ when $\sqrt{\log(1/\delta_n)} \geq \beta/2$. In such a case, we can obtain a $O\big(\delta_n^{\frac{2\beta}{2+\beta}}\big)$ rate even $\beta$ grows with $n$, as long as it grows slower than $\sqrt{\log(1/\delta_n)}$.

Our results for the iterative estimator match the state-of-the-art convergence rate with respect to $L_2$ norm for an iterative estimator in Bennett et al. (2023b). However, their method requires a minimax computation oracle, while our method does not.

# 9 NUMERICAL EXPERIMENTS

In this section, we evaluate our proposal by numerical simulation. In particular, we present the performance of RDIV when we use neural networks as the function approximator and the validity of the proposed model selection procedure. We show that with model selection, our method can achieve state-of-the-art performance in a wide range of data-generating processes.

## 9.1 EXPERIMENTAL SETTINGS

**Experiment Design.** In our experiment, we test our method on a synthetic dataset. We adjust the data generating process (DGP) for proximal causal inference used in Cui et al. (2020); Miao et al. (2018); Deaner (2021). Concretely, we generate multi-dimensional variables $U', S', W', Q', A$, where $U$ is an unobserved confounder, $S' \in d_S$ is the observed covariate, $W' \in d_W$ is the negative control outcomes, $Q' \in d_Q$ is the negative control actions, and $A$ is the selected treatment. We left the detailed generation process in Appendix J. For a detailed understanding of this setup, we refer the reader to Section 2 of Kallus et al. (2021). It is well known that there exists a bridge function $h'_0$ such that the following moment condition holds (Cui et al., 2020; Kallus et al., 2021):

$$\mathbb{E}[Y - h'_0(W', A, S')|Q', A, S'] = 0,$$

which allows the concrete form of equation 1. To introduce nonlinearity, we transform $(S', W', Q')$ into $(S, W, Q)$ via $S = g(S'), W = g(W'), Q = g(Q')$, where $g(\cdot)$ is a nonlinear invertible function applied elementwise to $S', W', Q'$ respectively. We consider several forms of $g(\cdot)$, including identity, polynomial, sigmoid design, and exponential function. In the final data, we only observe $(S, W, Q)$ but not $(S', W', Q')$. Here we use 6 different $g(\cdot)$: $\mathrm{Id}(t) = t$, $\mathrm{Poly}(t) = t^3$, $\mathrm{LogSigmoid}(t) = \log(1 + |16 * x - 8|) \cdot \mathrm{sign}(x)$, $\mathrm{Piecewise}(t) = 3(x-2)1_{x \leq 1} + \log(8x - 8)1_{x \geq 1}$, $\mathrm{Sigmoid}(t) = \frac{5}{1 + \exp(-0.1 * x)}$ and $\mathrm{CubicRoot} = x^{1/3}$.

**Methods to compare.** In this experiment, our goal is to estimate the counterfactual mean parameter $\mathbb{E}[Y(1)]$, which is unique as long as equation 1 holds. We learn $h_0$ in equation 1 by RDIV, which corresponds to the procedure in Algorithm 1 with MLE for conditional density estimation. We show results for different values for $\alpha \in \{0.01, 0.1\}$, and compare the performance of our approach to that of several different methods, including KernelIV (Singh et al., 2019), DeepIV (Hartford et al., 2017), DeepFeatureIV (Xu et al., 2021), and AGMM (Dikkala et al., 2020a). Note that DeepIV can be viewed as a special case of our methods, with $\alpha$ fixed to be 0. In the first stage of our algorithm, we use a three-layer mixture density network (Hartford et al., 2017; Rothfuss et al., 2019) as the approximator of the conditional density. In the second stage, we use a three-layer fully-connected neural network as the approximators for RDIV, DeepIV, AGMM, and DFIV. We present the results of our method and its comparison with previous benchmarks in terms of MSE normalized by the true estimand value in Table 2-5. Every estimate is calculated by 100 random replications. The confidence interval is calculated by 2 times the standard deviation.

## 9.2 RESULTS

First, we can observe that although our estimator resembles DeepIV, the later fix $\alpha = 0$ in equation 10, RDIV outperforms DeepIV for all $g(\cdot)$. This is due to the nonzero regularization term, which improves the performance of our estimator by a better tradeoff between bias and variance. Second, in most cases, AGMM and DFIV are outperformed by algorithms that only need single-level optimization (RDIV, KernelIV, DeepIV). This would be because, in these methods, optimization of the loss function is much harder, which results in the inaccuracy of estimators.

Table 2: $\mathbb{E}[Y(1)]$: $d_S = d_Q = 15, d_W = 1, n_1 = 500$.

| $g(t)$ | RDIV ($\alpha = 0.01$) | RDIV ($\alpha = 0.1$) | KernelIV | DeepIV | DFIV | AGMM |
|---|---|---|---|---|---|---|
| $\mathrm{Id}(t)$ | $0.0077 \pm 0.0012$ | $\mathbf{0.0021 \pm 0.0007}$ | $0.0193 \pm 0.0018$ | $0.0089 \pm 0.0015$ | $0.1069 \pm 0.0218$ | $0.0198 \pm 0.0011$ |
| $\mathrm{Poly}(t)$ | $\mathbf{0.0150 \pm 0.0057}$ | $0.0904 \pm 0.0202$ | $0.0439 \pm 0.0062$ | $0.0887 \pm 0.0276$ | $0.0920 \pm 0.0046$ | $0.0453 \pm 0.0023$ |
| $\mathrm{LogSigmoid}(t)$ | $0.0094 \pm 0.0013$ | $\mathbf{0.0022 \pm 0.0009}$ | $0.0031 \pm 0.0008$ | $0.0152 \pm 0.0026$ | $0.1444 \pm 0.0080$ | $0.0042 \pm 0.0010$ |
| $\mathrm{Piecewise}(t)$ | $0.0070 \pm 0.0017$ | $\mathbf{0.0024 \pm 0.0009}$ | $0.0041 \pm 0.0012$ | $0.0076 \pm 0.0012$ | $0.0150 \pm 0.0026$ | $0.0128 \pm 0.0024$ |
| $\mathrm{Sigmoid}(t)$ | $0.0206 \pm 0.0026$ | $\mathbf{0.0021 \pm 0.0006}$ | $0.0380 \pm 0.0025$ | $0.0278 \pm 0.0025$ | $0.1846 \pm 0.0092$ | $0.0070 \pm 0.0014$ |
| $\mathrm{CubicRoot}(t)$ | $0.0095 \pm 0.0014$ | $\mathbf{0.0024 \pm 0.0007}$ | $0.0511 \pm 0.0039$ | $0.0161 \pm 0.0018$ | $0.1357 \pm 0.0200$ | $0.0536 \pm 0.0021$ |

Table 3: $d_S = d_Q = 15, d_W = 1, n_1 = 1000$.

| $g(t)$ | RDIV ($\alpha = 0.01$) | RDIV ($\alpha = 0.1$) | KernelIV | DeepIV | DFIV | AGMM |
|---|---|---|---|---|---|---|
| Id$(t)$ | $0.0106 \pm 0.0013$ | $\mathbf{0.0014 \pm 0.0003}$ | $0.0145 \pm 0.0013$ | $0.0128 \pm 0.0015$ | $0.1162 \pm 0.0052$ | $0.0217 \pm 0.0135$ |
| Poly$(t)$ | $0.0164 \pm 0.0020$ | $\mathbf{0.0037 \pm 0.0027}$ | $0.0396 \pm 0.0038$ | $0.0182 \pm 0.0023$ | $0.1256 \pm 0.0044$ | $0.0054 \pm 0.0031$ |
| LogSigmoid$(t)$ | $0.0078 \pm 0.0009$ | $\mathbf{0.0009 \pm 0.0003}$ | $0.0259 \pm 0.0023$ | $0.0262 \pm 0.0023$ | $0.1618 \pm 0.0482$ | $0.0053 \pm 0.0010$ |
| Piecewise $(t)$ | $\mathbf{0.0017 \pm 0.0004}$ | $0.0059 \pm 0.0008$ | $0.0080 \pm 0.0008$ | $0.0019 \pm 0.0005$ | $0.1623 \pm 0.0674$ | $0.0014 \pm 0.0011$ |
| Sigmoid$(t)$ | $\mathbf{0.0077 \pm 0.0016}$ | $0.0082 \pm 0.0023$ | $0.0311 \pm 0.0014$ | $0.0110 \pm 0.0019$ | $0.2085 \pm 0.0443$ | $0.0296 \pm 0.0023$ |
| CubicRoot$(t)$ | $0.0254 \pm 0.0021$ | $\mathbf{0.0048 \pm 0.0008}$ | $0.0459 \pm 0.0024$ | $0.0248 \pm 0.0022$ | $0.1401 \pm 0.0047$ | $0.0650 \pm 0.0035$ |

Table 4: $d_S = d_Q = 20, d_W = 10, n_1 = 500$.

| $g(t)$ | RDIV ($\alpha = 0.01$) | RDIV ($\alpha = 0.1$) | KernelIV | DeepIV | DFIV | AGMM |
|---|---|---|---|---|---|---|
| Id$(t)$ | $0.0272 \pm 0.0022$ | $\mathbf{0.0055 \pm 0.0009}$ | $0.0088 \pm 0.0016$ | $0.0364 \pm 0.0025$ | $0.0291 \pm 0.0060$ | $0.3291 \pm 0.0115$ |
| Poly$(t)$ | $\mathbf{0.0067 \pm 0.0016}$ | $0.0230 \pm 0.0051$ | $0.0697 \pm 0.0041$ | $0.0263 \pm 0.0050$ | $0.0997 \pm 0.0046$ | $0.0409 \pm 0.0225$ |
| LogSigmoid$(t)$ | $0.0905 \pm 0.0058$ | $0.0525 \pm 0.0054$ | $0.0335 \pm 0.0014$ | $0.0960 \pm 0.0066$ | $0.2059 \pm 0.0826$ | $\mathbf{0.0218 \pm 0.0027}$ |
| Piecewise$(t)$ | $0.0305 \pm 0.0043$ | $\mathbf{0.0104 \pm 0.0021}$ | $0.0359 \pm 0.0010$ | $0.0225 \pm 0.0031$ | $0.7626 \pm 0.9996$ | $0.0136 \pm 0.0010$ |
| Sigmoid$(t)$ | $0.1481 \pm 0.0083$ | $0.0106 \pm 0.0028$ | $\mathbf{0.0018 \pm 0.0004}$ | $0.1983 \pm 0.0117$ | $0.3545 \pm 0.0494$ | $0.0307 \pm 0.0195$ |
| CubicRoot$(t)$ | $0.0810 \pm 0.0039$ | $0.0288 \pm 0.0025$ | $\mathbf{0.0021 \pm 0.0004}$ | $0.0949 \pm 0.0050$ | $0.0956 \pm 0.0453$ | $0.3461 \pm 0.0121$ |

Table 5: $d_S = d_Q = 20, d_W = 10, n_1 = 1000$.

| $g(t)$ | RDIV ($\alpha = 0.01$) | RDIV ($\alpha = 0.1$) | KernelIV | DeepIV | DFIV | AGMM |
|---|---|---|---|---|---|---|
| Id$(t)$ | $0.0652 \pm 0.0035$ | $0.0269 \pm 0.0020$ | $\mathbf{0.0009 \pm 0.0002}$ | $0.0639 \pm 0.0033$ | $0.1442 \pm 0.2461$ | $0.1321 \pm 0.0029$ |
| Poly$(t)$ | $0.0861 \pm 0.0076$ | $\mathbf{0.0224 \pm 0.0034}$ | $0.0465 \pm 0.0021$ | $0.1148 \pm 0.0082$ | $0.0951 \pm 0.0031$ | $0.1796 \pm 0.0023$ |
| LogSigmoid$(t)$ | $0.0649 \pm 0.0046$ | $0.0280 \pm 0.0025$ | $\mathbf{0.0197 \pm 0.0014}$ | $0.0759 \pm 0.0045$ | $0.2949 \pm 0.2917$ | $0.0247 \pm 0.0013$ |
| Piecewise$(t)$ | $0.0039 \pm 0.0008$ | $\mathbf{0.0037 \pm 0.0006}$ | $0.0215 \pm 0.0006$ | $0.0065 \pm 0.0012$ | $0.5442 \pm 0.4784$ | $0.0133 \pm 0.0009$ |
| Sigmoid$(t)$ | $0.1112 \pm 0.0053$ | $0.0091 \pm 0.0028$ | $\mathbf{0.0037 \pm 0.0005}$ | $0.1493 \pm 0.0058$ | $0.3332 \pm 0.0652$ | $0.0650 \pm 0.0029$ |
| CubicRoot$(t)$ | $0.0990 \pm 0.0042$ | $0.0802 \pm 0.0046$ | $\mathbf{0.0021 \pm 0.0004}$ | $0.1070 \pm 0.0043$ | $0.0956 \pm 0.0453$ | $0.3461 \pm 0.0121$ |

Table 6: Model selection results based on Best ERM. The left tabular is generated from a data size of $n_1 = 500$, while the right tabular is generated from a dataset with $n_1 = 1000$. Both datasets satisfies $d_S = d_Q = 20, d_W = 10$.

| $g(t)$ | RDIV ($\alpha = 0.01$) | RDIV ($\alpha = 0.1$) | KernelIV | RDIV ($\alpha = 0.01$) | RDIV ($\alpha = 0.1$) | KernelIV |
|---|---|---|---|---|---|---|
| Id$(t)$ | $\mathbf{0.0017 \pm 0.0017}$ | $0.0047 \pm 0.0021$ | $0.0088 \pm 0.0016$ | $0.0102 \pm 0.0028$ | $0.0014 \pm 0.0009$ | $\mathbf{0.0009 \pm 0.0002}$ |
| Poly$(t)$ | $\mathbf{0.0032 \pm 0.0024}$ | $0.0272 \pm 0.0097$ | $0.0697 \pm 0.0041$ | $0.0313 \pm 0.0137$ | $\mathbf{0.0049 \pm 0.0026}$ | $0.0465 \pm 0.0021$ |
| LogSigmoid$(t)$ | $0.0121 \pm 0.0055$ | $\mathbf{0.0019 \pm 0.0007}$ | $0.0335 \pm 0.0014$ | $0.0078 \pm 0.0020$ | $\mathbf{0.0008 \pm 0.0004}$ | $0.0197 \pm 0.0014$ |
| Piecewise$(t)$ | $0.0159 \pm 0.0121$ | $\mathbf{0.0020 \pm 0.0019}$ | $0.0359 \pm 0.0010$ | $\mathbf{0.0024 \pm 0.0013}$ | $0.0034 \pm 0.0027$ | $0.0215 \pm 0.0006$ |
| Sigmoid$(t)$ | $0.1655 \pm 0.0144$ | $0.0937 \pm 0.0174$ | $\mathbf{0.0018 \pm 0.0004}$ | $0.1538 \pm 0.0078$ | $0.0863 \pm 0.0187$ | $\mathbf{0.0037 \pm 0.0005}$ |
| CubicRoot$(t)$ | $0.0034 \pm 0.0017$ | $\mathbf{0.0019 \pm 0.0021}$ | $0.0021 \pm 0.0004$ | $0.0148 \pm 0.0048$ | $0.0036 \pm 0.0035$ | $\mathbf{0.0021 \pm 0.0004}$ |

Table 7: Model selection results based on Best ERM. Here $d_S = d_Q = 20, d_W = 10, n_1 = 500$

| $g(t)$ | RDIV ($\alpha = 0.01$) | RDIV ($\alpha = 0.1$) | KernelIV | DeepIV | DFIV | AGMM |
|---|---|---|---|---|---|---|
| Id$(t)$ | $\mathbf{0.0017 \pm 0.0017}$ | $0.0047 \pm 0.0021$ | $0.0088 \pm 0.0016$ | $0.0364 \pm 0.0025$ | $0.0291 \pm 0.0060$ | $0.3291 \pm 0.0115$ |
| Poly$(t)$ | $\mathbf{0.0032 \pm 0.0024}$ | $0.0272 \pm 0.0097$ | $0.0697 \pm 0.0041$ | $0.0313 \pm 0.0137$ | $0.0997 \pm 0.0046$ | $0.0409 \pm 0.0225$ |
| LogSigmoid$(t)$ | $0.0121 \pm 0.0055$ | $\mathbf{0.0019 \pm 0.0007}$ | $0.0335 \pm 0.0014$ | $0.0960 \pm 0.0066$ | $0.2059 \pm 0.0826$ | $0.0218 \pm 0.0027$ |
| Piecewise$(t)$ | $0.0159 \pm 0.0121$ | $\mathbf{0.0020 \pm 0.0019}$ | $0.0359 \pm 0.0010$ | $0.0225 \pm 0.0031$ | $0.7626 \pm 0.9996$ | $0.0136 \pm 0.0010$ |
| Sigmoid$(t)$ | $0.1655 \pm 0.0144$ | $0.0937 \pm 0.0174$ | $\mathbf{0.0018 \pm 0.0004}$ | $0.1983 \pm 0.0117$ | $0.3545 \pm 0.0494$ | $0.0307 \pm 0.0195$ |
| CubicRoot$(t)$ | $0.0034 \pm 0.0017$ | $\mathbf{0.0019 \pm 0.0021}$ | $0.0021 \pm 0.0004$ | $0.0949 \pm 0.0050$ | $0.0956 \pm 0.0453$ | $0.3461 \pm 0.0121$ |

## 10 CONCLUSION

In this paper, we study NPIV regression with general function approximation. We analyze a Tikhonov-regularized variant of the well-established DeepIV estimator, namely the Regularized DeepIV (RDIV). We show that our estimator converges to the least norm solution, and derive its convergence rate. Notably, we prove that such an estimator does not rely on uniqueness or minimax computation oracle. We further illustrate that RDIV can be incorporated into model selection and show that our procedure can achieve the oracle rate with respect to the minimal model misspecification error. When extended to an iterative estimator, RDIV achieves a state-of-the-art convergence rate. Moreover, we justify our method through numerical simulations. Our experiments show that RDIV outperforms existing benchmarks in a wide range of circumstances.

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

## A  RELATED WORKS

**Nonparametric IV problem.**    Nonparametric IV estimation has been extensively explored in past decades. Such estimation is tough to solve even when both the linear operator $\mathcal{T}$ and the response $r_0$ are known, known as ill-posedness. The ill-posedness often refers to the presence of one or more of the following characteristics: (1) the absence of solutions, (2) the existence of multiple solutions, and (3) the discontinuity of the inverse of operator $\mathcal{T}$. Many traditional nonparametric estimators have been proposed to address these challenges, such as series-based estimators (Florens et al., 2011; Ai and Chen, 2003; Chen, 2021; Chen and Pouzo, 2012; Darolles et al., 2011) and kernel-based estimators (Hall and Horowitz, 2005; Horowitz, 2007; Singh et al., 2019). However, these methods cannot directly accommodate modern machine-learning techniques like neural networks with theoretical soundness.

Recently, there has been growing interest in the application of general function approximation techniques, such as deep neural networks and random forests, to IV problems in a unified manner. Among those methods, Bennett and Kallus (2020); Dikkala et al. (2020b); Lewis and Syrgkanis (2018); Liao et al. (2020a); Zhang et al. (2023) reformulate the conditional moment constraint into a minimax optimization and use its solution as the estimator. Notably, Liao et al. (2020a); Bennett et al. (2023b;a) establish $L_2$ convergence by linking minimax optimization with Tikhonov regularization under the assumption of the source condition. Moreover, Liao et al. (2020b) assumes uniqueness of solution $h_0$. Dikkala et al. (2020b); Lewis and Syrgkanis (2018) provide a guarantee for the projected MSE without further assumptions. However, they could not guarantee the convergence rate in strong $L_2$ metric when multiple solutions to conditional moment constraint exist. Furthermore, these methods require a computation oracle for minimax optimization, which further makes model selection challenging. In contrast, our method does not require computational oracles and enables model selection with statistical guarantees.

Several existing works eschew the need for minimax optimization oracles (Chen and Pouzo, 2012; Hartford et al., 2017; Xu et al., 2021). As the most related work, DeepIV (Hartford et al., 2017) introduces a similar loss function to us. However, it lacks an explicit regularization term, which results in the lack of theoretical guarantee and the lack of guarantee for model selection. As another work, Xu et al. (2021) extends the two-stage kernel algorithm in Singh et al. (2019) to deep neural networks, but their algorithm is essentially a bilevel optimization problem, which is hard to solve in general (Hong et al., 2023; Khanduri et al., 2021; Guo et al., 2021). Notably, Chen and Pouzo (2012) considers a more general conditional moment restriction of $\mathbb{E}[\rho(X, h_0) \mid Z] = 0$ and obtains an estimator of $h_0$ by minimizing a penalized sieve minimum distance (PSMD). Their method is similar to ours: first, they assume the existence of an estimate $\hat{m}(h, Z)$ for $m(h, Z) = \mathbb{E}[h(X) \mid Z]$ for all $h \in L_2(X)$. Then they minimize $\mathbb{E}_n[\|\hat{m}(h, Z)\|2] + \lambda_n \hat{P}_n(h)$ over all possible $h \in \mathcal{H}$, where $\mathcal{H}$ is a sieve space with a growing dimension, and $\hat{P}_n$ is a nonrandom penalty function. However, their theory is limited to the case when $h_0$ is identifiable and is well approximated by the sieve estimator, therefore can not be straightforwardly generalized to general function approximation with model misspecification. Recently, Chen et al. (2024) proposed a TOSG method under the lens of optimization, and established its convergence rate for including linear function class and general linear function class with a known link function $g$. However, their techniques are not directly transferable to general function approximation, and no guarantee or procedure for model selection for $g$ is discussed. Moreover, Chen et al. (2024) assumes a two-sample oracle that outputs two independently sampled $X$ and $X'$ conditioned on the same instrument $Z$, which is often hard to satisfy in practice.

**Model selection.**    Model selection has been well studied in the regression and supervised machine learning literature (Bartlett et al., 2002a; Gold and Sollich, 2003; McAllester, 2003). The objective can be described more concretely as follows: given $M$ candidate models, $\{f_1, \ldots, f_M\}$, each having some statistical complexity $\delta_j$ and some approximation error $\epsilon_j$ (with respect to some un-known true model $f_0$) we wish to find an aggregated model $\hat{f}$ whose mean squared error is closed to the optimal trade-off between statistical complexity and approximation error among all models, i.e.: $\|\hat{f} - f_0\| \lesssim \min_{j=1}^M \delta_j + \epsilon_j$. The statistical complexity of a function space can be accurately characterized, albeit the approximation error is un-attainable as it relates to the unknown true model. A guarantee of the form above implies that using the observed data we can compete (up to constants) with an oracle that knows the approximation errors and chooses the best model space. We leave the detailed summary of

existing works in Appendix B. Despite the abundance of methodologies for IV regression problems, few studies have investigated model misspecification and provided model selection procedures to select the best model class. As a few exceptional works, while Xu et al. (2021) and Ai and Chen (2007) considered the misspecified regime, but they did not discuss model selection approaches. A typical approach to model selection is out-of-sample validation: estimate different models on half the data and select the estimated model that achieves the smallest empirical risk on the second half (or the best convex ensemble of models that achieves the smallest out-of-sample risk). One problem that arises for model selection in this IV regression setup is to transform the excess risk guarantees, which will be in terms of the weak metric, i.e. $\|\mathcal{T}(\cdot)\|_2$, into the desired bound in the $L_2$ error. In this work, we show that by leveraging the Tikhonov regularization, we can achieve an MSE bound that achieves the same order as the oracle function class.

## B  ADDITIONAL RELATED WORKS FOR MODEL SELECTION

**Model Selection.**   Under the classical supervised learning setting, a common approach is to perform empirical risk minimization (ERM) on a separate validation set, and choose the candidate model that achieves the smallest risk (Mitchell and van de Geer, 2009), or similarly, through M-fold cross-validation which splits the data into M folds, and evaluates the risk on the different held out set for each model (Vaart et al., 2006). As an alternative to selecting a single model, convex aggregation or linear aggregation is employed to find the best convex/linear combination of models (Lecué, 2013; Lecué and Mendelson, 2014). However, it can be shown that the aforementioned approaches are sub-optimal in the sense that they cannot achieve the optimal $\frac{\log(M)}{n}$ rate for the model selection residual. To tackle this challenge, Lecué and Mendelson (2009) proposed a different approach for convex aggregation by first finding a subset of "almost minimizers" - a subset of the candidate functions that is sufficiently close to the minimizer within the candidates on the validation set, and then finding a best aggregate in the convex hull of this subset. This approach achieves the optimal model selection rates as it performs ERM on a subset that is much smaller than the convex hull of all candidate models, thereby reducing the statistical error. Furthermore, other optimal model selection approaches include the Q-aggregation approach which performs ERM with a modified loss that adds an additional penalty based on individual model performance (Lecué and Rigollet, 2014).

## C  RESULTS WHEN USING $\chi^2$-MLE

In this section, we consider another density estimation for the density estimation, the $\chi^2$-MLE:

$$\hat{g} = \arg\min_{g \in \mathcal{G}} 0.5 \cdot \mathbb{E}_n\left[\int_{\mathcal{X}} g^2(x|Z)d\mu(x)\right] - \mathbb{E}_n[g(X|Z)]. \tag{11}$$

### C.1  FINITE SAMPLE RESULTS

Although Assumption 4 is widely accepted in previous works, in practice, it often fails to hold when $g_0$ does not have full support on $\mathcal{X}$. To address this drawback of MLE, in this subsection, we further discuss the finite sample convergence rate of Algorithm 3 when the conditional density estimation is performed by $\chi^2$-MLE. In this case, the first step estimation procedure is given by Equation equation 11. Notably, our guarantee does not relate to the lower bound of $g_0(x|z)$. Our results rely on the following assumption, which characterizes the smoothness of function class $\mathcal{H}$.

**Assumption 9** ($\gamma$-Smoothness). *For all $h - h' \in \mathcal{H} - \mathcal{H}$, we assume that $\|h - h'\|_\infty \leq \|h - h'\|_2^\gamma$.*

Such a relationship is known for instance to hold for Sobolev spaces and more generally for reproducing kernel Hilbert spaces (RKHS) with a polynomial eigendecay. A notable instance is RKHS with eigendevay at a rate of $O(1/j^{1/p})$ for some $p \in (0, 1)$. In that case, Lemma 5.1 of Mendelson and Neeman (2010) shows that $\gamma = 1 - p$. For the Gaussian kernel, which has an exponential eigendecay, we can take $p$ arbitrarily close to 0. We now summarize our result for $\chi^2$-MLE in the following theorem.

**Theorem 10** ($L_2$ convergence rate for RMIV with $\chi^2$-MLE)**.** *Suppose Assumption 2,3,9 hold. By setting $\alpha = \delta_n^{\frac{2}{2+(2-\gamma)\min\{\beta,2\}}}$, with probability at least $1 - c_1\exp(c_2 n\delta_n^2)$, we have*

$$\|\hat{h} - h_0\|_2^2 \leq O\big(\delta_n^{\frac{2\min\{\beta,2\}}{2+(2-\gamma)\min\{\beta,2\}}}\big).$$

*Here $\delta_n$ has the same definition in Theorem 5.*

The convergence rate of RMIV with $\chi^2$-MLE depends on the smoothness parameter $\gamma$. As $\gamma \to 1$, we have $\|\hat{h} - h_0\|_2^2 \leq O\big(\delta_n^{\frac{2\min\{\beta,2\}}{2+\min\{\beta,2\}}}\big)$, which recovers the rate in Theorem 8. We further discuss the results for $\chi^2$-MLE based IV regression under misspecification.

**Theorem 11** ($L_2$ convergence rate for RMIV with $\chi^2$-MLE under misspecification)**.** *Suppose Assumption 2,9 hold, and there exists $h^\dagger \in \mathcal{H}$ and $g^\dagger \in \mathcal{G}$ such that $\|h_0 - h^\dagger\|_2 \leq \epsilon_{\mathcal{H}}$ and $\mathbb{E}\big[\int_{\mathcal{X}}(g^\dagger(x|Z) - g_0(x|Z))^2 d\mu(x)\big] \leq \epsilon_{\mathcal{G}}$. For any $0 < \alpha \leq 1$, with probability at least $1 - c_1\exp(c_2 n\delta_n^2)$, we have*

$$\|\hat{h} - h_0\|_2^2 \leq O\bigg(\bigg(\frac{\delta_n^2 + \epsilon_{\mathcal{G}}}{\alpha^2}\bigg)^{1/(2-\gamma)} + \alpha^{\min\{\beta+1,2\}-1} + \frac{\epsilon_{\mathcal{H}}^2}{\alpha}\bigg),$$

*Here $\delta_n$ has the same definition in Theorem 5.*

**Remark 3.** *We define $\epsilon := \{\epsilon_{\mathcal{G}}, \epsilon_{\mathcal{H}}^2\}$. If $\epsilon < 1$, then by setting $\alpha = (\delta_n^2 + \epsilon)^{\frac{2}{2+(2-\gamma)\min\{\beta,1\}}}$, we have*

$$\|\hat{h} - h_0\|_2^2 \leq O\big((\delta_n^2 + \epsilon)^{\frac{2\min\{\beta,1\}}{2+(2-\gamma)\min\{\beta,1\}}}\big).$$

*If $\epsilon \geq 1$, then by setting $\alpha = 1$, we have $\|\hat{h} - h_0\|_2^2 \leq O(\epsilon^{1/(2-\gamma)})$.*

## C.2 RESULTS FOR MODEL SELECTION

Theorem 7 is extended when using $\chi^2$-MLE. Indeed, if Assumption 9 holds and the candidate function are trained with $\hat{g}$ estimated using the $\chi^2$-MLE approach, the output of Convex-ERM or Best-ERM $\hat{\theta}$, satisfies

$$\|h_{\hat{\theta}} - h_0\|_2^2 \leq \min_j O\left(\alpha^{\min\{\beta+1,2\}-1} + \bigg(\frac{\delta_{n,j}^2 + \epsilon_{\mathcal{G}}}{\alpha^2}\bigg)^{1/(2-\gamma)} + \frac{1}{\alpha}\epsilon_{\mathcal{H}_j}^2\right).$$

## C.3 CONVERGENCE RESULTS FOR ITERATIVE VERSION

We further discuss the finite sample convergence rate of Algorithm 3 when the conditional density estimation is performed by $\chi^2$-MLE. In this case, the first step estimation procedure is given by Equation equation 11. Notably, in this case, we do not require the ground truth density $g_0$ to be uniformly lower bounded, which is assumed in Assumption 4 and serves as a prerequisite for MLE convergence. Our results are summarized by the following theorem.

**Theorem 12** ($L_2$ convergence rate for iterative $\chi^2$-MLE estimator)**.** *Under Assumption 1,2,3,9, by setting $\alpha = \delta_n^{\frac{2}{2+(2-\gamma)\min\{\beta,2m\}}}$, with probability at least $1 - c_1 m\exp(c_2 n\delta_n^2)$, we have*

$$\|\hat{h}_m - h_0\|_2 \leq O\big(16^{2m} \cdot \delta_n^{\frac{2\min\{\beta,2m\}}{2+(2-\gamma)\min\{\beta,2m\}}}\big).$$

*Here $\delta_n$ has the same definition in Theorem 5.*

**Remark 4.** *Similar to Section 6, by setting the iteration number $m = \lceil\min\{\beta/2, \log\log(1/\delta_n)\}\rceil$, we have*

$$\|\hat{h}_m - h_0\|_2 \leq O\bigg(16^{2m} \cdot \delta_n^{\frac{2\min\{\beta,2m\}}{2+(2-\gamma)\min\{\beta,2m\}}}\bigg).$$

*Therefore, for $\log\log\delta_n \geq \beta$, eventually we have the rate of $O\big(\delta_n^{\frac{2\beta}{2+(2-\gamma)\beta}}\big)$. If $\delta_n = O(n^{-\iota})$, then we can set $m = \lceil\min\{\beta/2, \sqrt{\log(1/\delta_n)}\}\rceil$ to obtain the same rate. Moreover, if $\gamma \to 1$, e.g. RKHS with exponential eigenvalue decay (Mendelson and Neeman, 2010, Lemma 5.1), then we recover the rate of $O\big(\delta_n^{\frac{2\beta}{2+\beta}}\big)$ even without Assumption 4.*

# D PROOF OF THEOREM 5 AND 10

In this section, we prove the convergence rate of non-iterative RMIV. We prove the results of Theorem 5 and 10 respectively. Recall that we define

$$h_* := \operatorname*{arg\,min}_{h \in \mathcal{H}} \|Y - \mathcal{T}h\|_2^2 + \alpha \|h\|_2^2, \tag{12}$$

by Lemma 6, we have

$$\|h_* - h_0\|_2^2 \le \|w_0\|^2 \alpha^{\min\{\beta, 2\}}.$$

Therefore, we only need to provide an upper bound for $\|\hat{h} - h_*\|_2^2$. We start by proving the following lemma, and with the convergence rate of MLE and $\chi^2$-MLE, we conclude the proof of Theorem 5 and Theorem 10 respectively.

**Lemma 1.** *With probability at least $1 - c_1 \exp(c_2 n \delta_{n,\mathcal{H}}^2)$, we have the following inequality:*

$$\alpha \|\hat{h} - h_*\|_2^2 + \|\mathcal{T}(\hat{h} - h_*)\|_2^2 \le \mathbb{E}[\ell_{\hat{h}, g_0} - \ell_{h_*, g_0}]$$

$$= O\left(\delta_{n,\mathcal{H}}\{(\alpha + 1)\|\hat{h} - h_*\|_2 + \delta_{n,\mathcal{H}}\} + \|(\hat{\mathcal{T}} - \mathcal{T})(\hat{h} - h_*)\|_1\right).$$

*Proof.* By the optimality of $h_*$ in Eq. equation 3, we have

$$\alpha \|\hat{h} - h_*\|_2^2 + \|\mathcal{T}(\hat{h} - h_*)\|_2^2 \le \mathbb{E}[L(\mathcal{T}\hat{h})] - \mathbb{E}[L(\mathcal{T}h_*)] + \alpha \{\mathbb{E}[\hat{h}^2(X)] - \mathbb{E}[h_*(X)^2]\},$$

where define $L(\mathcal{T}h) := (Y - \mathcal{T}h)^2$. Recall that

$$\mathbb{E}[L(\mathcal{T}\hat{h})] - \mathbb{E}[L(\mathcal{T}h_*)] + \alpha \{\mathbb{E}[\hat{h}^2(X)] - \mathbb{E}[h_*(X)^2]\} =$$

$$\mathbb{E}[-2Y\mathcal{T}(\hat{h} - h_*)(Z) + (\mathcal{T}\hat{h})^2(Z) - (\mathcal{T}h_*)^2(Z)] + \alpha \{\mathbb{E}[\hat{h}^2(X)] - \mathbb{E}[h_*(X)^2]\},$$

we have

$$\alpha \|\hat{h} - h_*\|_2^2 + \|\mathcal{T}(\hat{h} - h_*)\|_2^2$$

$$= \mathbb{E}[-2Y\mathcal{T}(\hat{h} - h_*)(Z) + (\mathcal{T}\hat{h})^2(Z) - (\mathcal{T}h_*)^2(Z)] + \alpha \{\mathbb{E}[\hat{h}^2(Z)] - \mathbb{E}[h_*(Z)^2]\}$$

$$= \mathbb{E}[-2Y\hat{\mathcal{T}}(\hat{h} - h_*)(Z) + (\hat{\mathcal{T}}\hat{h})^2(Z) - (\hat{\mathcal{T}}h_*)^2(Z)] + C_1 \times \mathbb{E}[|(\hat{\mathcal{T}} - \mathcal{T})(\hat{h} - h_*)(X)|]$$

$$\qquad + \alpha \{\mathbb{E}[\hat{h}^2(X)] - \mathbb{E}[h_*(X)^2]\}$$

$$\le \text{Emp} + \text{Loss} + C_1 \times \mathbb{E}[|(\hat{\mathcal{T}} - \mathcal{T})(\hat{h} - h_*)(Z)|]$$

$$= \text{Emp} + \text{Loss} + \|(\hat{\mathcal{T}} - \mathcal{T})(\hat{h} - h_*)\|_1, \tag{13}$$

here the inequality comes from the uniform boundedness of $\hat{h}, h_*, \mathcal{T}h, \mathcal{T}\hat{h}, \hat{\mathcal{T}}h, \hat{\mathcal{T}}\hat{h}$, and the $O(1)$-Lipschitz of $L(\cdot)$.

$$\text{Emp} = |(\mathbb{E}_n - \mathbb{E})[L(\hat{\mathcal{T}}\hat{h}) - L(\hat{\mathcal{T}}h_*) + \alpha(\hat{h}^2(X) - h_*(X)^2)]|,$$

$$\text{Loss} = \mathbb{E}_n[-2Y\hat{\mathcal{T}}(\hat{h} - h_*)(Z) + (\hat{\mathcal{T}}\hat{h})^2(Z) - (\hat{\mathcal{T}}h_*)^2(Z) + \alpha\{\hat{h}^2(X) - h_*(X)^2\}].$$

Here, using Lemma 4, the term Emp is upper-bounded as follows with probability at least $1 - c_1 \exp(c_2 n \delta_{n,\mathcal{H}}^2)$:

$$\text{Emp} \le \delta_{n,\mathcal{H}}\{\alpha \|\hat{h} - h_*\|_2 + \|\hat{\mathcal{T}}(\hat{h} - h_*)\|_2 + \delta_{n,\mathcal{H}}\}$$

$$\le \delta_{n,\mathcal{H}}\{\alpha \|\hat{h} - h_*\|_2 + \|\hat{h} - h_*\|_2 + \delta_{n,\mathcal{H}}\}. \tag{14}$$

Furthermore, recall that by our iteration in equation 5, we have

$$\mathbb{E}_n[-2Y\mathcal{T}(\hat{h} - h_*)(Z) + (\mathcal{T}\hat{h})^2(Z) - (\mathcal{T}h_*)^2(Z) + \alpha\{\hat{h}^2(X) - h_*(X)^2\}] \le 0.$$

Hence, we have

$$\text{Loss} \le 0. \tag{15}$$

Combining everything, we have

$$\alpha \|\hat{h} - h_*\|_2^2 + \|\mathcal{T}(\hat{h} - h_*)\|_2^2 \le \delta_{n,\mathcal{H}}\{(\alpha + 1)\|\hat{h} - h_*\|_2 + \delta_{n,\mathcal{H}}\} + \|(\hat{\mathcal{T}} - \mathcal{T})(\hat{h} - h_*)\|_1, \tag{16}$$

Here the constant $c_1$ and $c_2$ hide constants related to $C, C_0$. The first inequality comes from equation 14. We implicitly use $\alpha \le 1$ in the last inequality. □

**Proof of Theorem 5.** By Assumption 3, we have $\epsilon_{\mathcal{G}} = 0$. By Corollary 1 and Lemma 1, since $\alpha \leq 1$ we have

$$\alpha\|\hat{h} - h_*\|_2^2 + \|\mathcal{T}(\hat{h} - h_*)\|_2^2 = O\left(\delta_{n,\mathcal{H}}\{(\alpha+1)\|\hat{h} - h_*\|_2 + \delta_{n,\mathcal{H}}\} + \delta_{n,\mathcal{G}}\|\hat{h} - h_*\|_2\right)$$

$$\leq c_1\delta_n^2 + c_2\delta_n\|\hat{h} - h_*\|_2 \qquad (\delta_n := \max\{\delta_{n,\mathcal{G}}, \delta_{n,\mathcal{H}}\})$$

$$\leq c_1\delta_n^2 + 2c_2''\delta_n^2/\alpha + 2c_2'\alpha\|\hat{h} - h^*\|_2^2 \qquad (2ab \leq ca^2 + \frac{b^2}{c})$$

holds with probability at least $1 - c\exp(n\delta_n^2)$, where $c_2' \leq 1$. By Lemma 7, we have

$$\|\hat{h} - h_*\|_2^2 \leq O((\delta_n^2/\alpha^2) + \delta_n^2/\alpha) = O(\delta_n^2/\alpha^2), \qquad (\alpha \leq 1)$$

therefore by Lemma 6, we have

$$\|\hat{h} - h_0\|_2^2 \leq \delta_n^2/\alpha^2 + \alpha^{\min(\beta,2)},$$

set $\alpha = \delta_n^{\frac{2}{2+\min\{\beta,2\}}}$, and we conclude the proof of Theorem 5.

**Proof of Theorem 10.** By Assumption 3, we have $\epsilon_{\mathcal{G}} = 0$. By Corollary 1 and Lemma 1, we have

$$\alpha\|\hat{h} - h_*\|_2^2 + \|\mathcal{T}(\hat{h} - h_*)\|_2^2 \leq \delta_{n,\mathcal{H}}\{\alpha\|\hat{h} - h_*\|_2 + \|\mathcal{T}(h - h_*)\|_2 + \delta_{n,\mathcal{G}}\|\hat{h} - h_*\|_\infty + \delta_{n,\mathcal{H}}\} + \delta_{n,\mathcal{G}}\|\hat{h} - h_*\|_\infty,$$

By Assumption 9, we have

$$\alpha\|\hat{h} - h_*\|_2^2 + \|\mathcal{T}(\hat{h} - h_*)\|_2^2 \leq \delta_{n,\mathcal{H}}\{(\alpha+1)\|\hat{h} - h_*\|_2 + \delta_{n,\mathcal{H}}\} + \delta_{n,\mathcal{G}}\|\hat{h} - h_*\|_2^\gamma$$

$$\leq c_1\delta_n\|\hat{h} - h_*\|_2 + c_2\delta_n\|\hat{h} - h_*\|_2^\gamma, \qquad (\delta_n := \{\delta_{n,\mathcal{G}}, \delta_{n,\mathcal{H}}\})$$

By Lemma 7, we have

$$\|\hat{h} - h_*\|_2^2 \leq O((\delta_n/\alpha)^{\frac{2}{2-\gamma}} + (\delta_n/\alpha)^2) \leq O(\delta_n/\alpha)^{\frac{2}{2-\gamma}}$$

since $\gamma \in (0, 1)$. Therefore, by Lemma 6, we have

$$\|\hat{h} - h_0\|_2^2 \leq (\delta_n/\alpha)^{\frac{2}{2-\gamma}} + \alpha^{\min(\beta,2)}.$$

By selecting $\alpha = O(\delta_n^{\frac{2}{2+(2-\gamma)\min\{\beta,2\}}})$, we have

$$\|\hat{h} - h_0\|_2^2 \leq \delta_n^{\frac{2\min\{\beta,2\}}{2+(2-\gamma)\min\{\beta,2\}}},$$

and we conclude the proof of Theorem 10.

# E    PROOF OF THEOREM 6 AND 11

In this section, we consider the case when $\epsilon_{\mathcal{G}}$ and $\epsilon_{\mathcal{H}}$ doht equal zero, i.e. Assumption 3 does not hold. We aim to establish a convergence rate for $\|\hat{h} - h_0\|_2$ for both MLE-based RDIV and $\chi^2$-MLE based RDIV in terms of $\delta_n$, $\epsilon_{\mathcal{H}}$ and $\epsilon_{\mathcal{G}}$.

**Lemma 2.** *Under Assumption 2, for $\alpha \in (0, 1)$ we have*

$$\|\hat{h} - h_0\|^2 \leq 3\|\hat{h} - h_*\|^2 + O\left(\frac{1}{\alpha}\left\{\epsilon_{\mathcal{H}}^2 + \alpha^{\min\{\beta+1,2\}}\right\}\right),$$

*Proof.* Note that in the misspecified case, we no longer have $h_0 \in \mathcal{H}$. We further a augmented function class $\mathcal{H}' = \text{Span}(\mathcal{H} \cup \{h_0\})$, and the corresponding optimizer of $\mathcal{L}_0$ on $\mathcal{H}$ and $\mathcal{H}'$:

$$h_*' = \underset{h \in \mathcal{H}'}{\arg\min} \|\mathcal{T}(h - h_0)\|_2^2 + \alpha\|h\|^2,$$

$$h_* = \underset{h \in \mathcal{H}}{\arg\min} \|\mathcal{T}(h - h_0)\|_2^2 + \alpha\|h\|^2.$$

We define a function

$$\mathcal{L}_0(t) := \|\mathcal{T}(h_*' + t(h_* - h_*') - h_0)\|_2^2 + \alpha\|h_*' + t(h_* - h_*')\|^2,$$

then $\mathcal{L}_0$ is $\alpha$-strongly convex, and attains its minimum at $\mathcal{L}_0(0)$. Note that we have the following inequality holds for all $h \in \mathcal{H}$,

$$\frac{1}{\alpha}(L_0(1) - L_0(0)) = \frac{1}{\alpha}\left\{ \|\mathcal{T}(h_* - h_0)\|^2 + \alpha\|h_*\|^2 - (\|\mathcal{T}(h_*' - h_0)\|^2 + \alpha\|h_*'\|^2) \right\}$$

$$\leq \frac{1}{\alpha}\left\{ \|\mathcal{T}(h - h_0)\|^2 + \alpha\|h\|^2 - (\|\mathcal{T}(h_*' - h_0)\|^2 + \alpha\|h_*'\|^2) \right\}$$

$$\text{(Optimality of } h_*')$$

$$= \frac{1}{\alpha}\{\|\mathcal{T}(h - h_*')\|^2 + \alpha\|h_*'\|^2\} \qquad \text{(First order condition of } h_*')$$

$$\leq \frac{2}{\alpha}\left\{ 2\|\mathcal{T}(h - h_0)\|^2 + 2\|\mathcal{T}(h_*' - h_0)\|^2 + 2\alpha\|h - h_0\|^2 + 2\alpha\|h_*' - h_0\|^2 \right\}$$

$$\leq \frac{2}{\alpha}\{4\|h - h_0\|^2 + O(\|w_0\|^2 \alpha^{\min\{\beta+1,2\}})\},$$

set $h = h^\dagger$, by strong convexity and $\partial\mathcal{L}_0(0) = 0$, we have

$$\|h_* - h_*'\|^2 \leq \frac{1}{\alpha}|\mathcal{L}_0(1) - \mathcal{L}(0)| \leq O(\frac{1}{\alpha}\{\epsilon_{\mathcal{H}}^2 + \alpha^{\min\{\beta+1,2\}}\}).$$

Therefore we have

$$\|\hat{h} - h_0\|^2 \leq 3\left\{ \|\hat{h} - h_*\|^2 + \|h_* - h_*'\|^2 + \|h_*' - h_0\|^2 \right\}$$

$$= 3\|\hat{h} - h_*\|^2 + O\left( \frac{1}{\alpha}\left\{ \epsilon_{\mathcal{H}}^2 + \alpha^{\min\{\beta+1,2\}} \right\} \right) + 3\alpha^{\min\{\beta,2\}},$$

and we conclude our proof for the lemma. $\qquad\square$

**Proof for Theorem 6.** By Lemma 1, we have

$$\alpha\|\hat{h} - h_*\|^2 = O\left( \delta_{n,\mathcal{H}}\left\{ (\alpha+1)\|\hat{h} - h_*\|_2 + \delta_{n,\mathcal{H}} \right\} + \|(\hat{\mathcal{T}} - \mathcal{T})(\hat{h} - h_*)\|_1 \right),$$

By Corollary 1, we have $\|(\mathcal{T} - \hat{\mathcal{T}})(\hat{h} - h_*)\|_1 \leq (\delta_{n,\mathcal{G}}^2 + \epsilon_{\mathcal{G}})^{1/2}\|\hat{h} - h_*\|$, and we have

$$\|\hat{h} - h_*\|^2 \leq \frac{1}{\alpha} \cdot O\left( \delta_{n,\mathcal{H}}\|\hat{h} - h_*\| + (\delta_{n,\mathcal{G}}^2 + \epsilon_{\mathcal{G}})^{1/2}\|\hat{h} - h_*\| + \delta_{n,\mathcal{G}}^2 \right),$$

therefore by Lemma 7, we have

$$\|\hat{h} - h_*\|^2 = O\left( \frac{\delta_{n,\mathcal{G}}^2 + \epsilon_{\mathcal{G}} + \delta_{n,\mathcal{H}}^2}{\alpha^2} \right). \tag{17}$$

By Lemma 2, combine everything together:

$$\|\hat{h} - h_0\|^2 = O\left( \frac{\delta_{n,\mathcal{G}}^2 + \delta_{n,\mathcal{H}}^2 + \epsilon_{\mathcal{G}}}{\alpha^2} + \alpha^{\min\{\beta+1,2\}-1} + \frac{\epsilon_{\mathcal{H}}^2}{\alpha} \right).$$

note that $\delta_n := \{\delta_{n,\mathcal{G}}, \delta_{n,\mathcal{H}}\}$, we conclude the proof of Theorem 6.

**Proof of Theorem 11.** By Lemma 1, we have

$$\alpha\|\hat{h} - h_*\|^2 \leq O\left( \delta_n\left\{ (\alpha+1)\|\hat{h} - h_*\|_2 + \delta_n \right\} + \|(\hat{\mathcal{T}} - \mathcal{T})(\hat{h} - h_*)\|_1 \right)$$

$$\leq O\left( \delta_n\left\{ (\alpha+1)\|\hat{h} - h_*\|_2 + \delta_n \right\} + \|(\hat{\mathcal{T}} - \mathcal{T})(\hat{h} - h_*)\|_2 \right),$$

by Lemma 2, we have $\|(\hat{\mathcal{T}} - \mathcal{T})(\hat{h} - h_*)\|_2 \leq (\delta_n^2 + \epsilon_{\mathcal{G}})^{1/2}\|\hat{h} - h_*\|_\infty$, therefore we have

$$\|\hat{h} - h_*\|^2 \leq \frac{1}{\alpha} \cdot O\bigg(\delta_n\|\hat{h} - h_*\| + (\delta_n^2 + \epsilon_{\mathcal{G}})^{1/2}\|\hat{h} - h_*\|_\infty\bigg) \qquad (\alpha \leq 1)$$

$$\leq \frac{1}{\alpha} \cdot O\bigg(\delta_n\|\hat{h} - h_*\| + (\delta_n^2 + \epsilon_{\mathcal{G}})^{1/2}\|\hat{h} - h_*\|_2^\gamma\bigg),$$

where the second inequality comes from Assumption 9. By Lemma 7, we have

$$\|\hat{h} - h_*\|^2 \leq O\bigg(\bigg(\frac{\delta_n^2 + \epsilon_{\mathcal{G}}}{\alpha^2}\bigg)^{1/(2-\gamma)}\bigg) \qquad (18)$$

by Lemma 2, combine everything together, we have

$$\|\hat{h} - h_0\|^2 \leq O\bigg(\bigg(\frac{\delta_n^2 + \epsilon_{\mathcal{G}}}{\alpha^2}\bigg)^{1/(2-\gamma)} + \alpha^{\min\{\beta+1,2\}-1} + \frac{\epsilon_{\mathcal{H}}^2}{\alpha}\bigg),$$

and thus we conclude the proof of Theorem 11.

## F    PROOF OF THEOREM 7

In this section, we will provide the details for the model selection results in the paper. Let $\ell_{h,g}(Y, Z, X)$ denote the loss evaluated for a function $h$ using the likelihood function $\hat{g}$:

$$\ell_{h,\hat{g}}(Y, Z, X) = \bigg(Y - \int h(x)\hat{g}(x|Z)\mu(dx)\bigg)^2 + \alpha h(X)^2$$

Also, to simplify the notation, we use $\{X_i, Y_i, Z_i\}$ instead of $\{X_i', Y_i', Z_i'\}$.

For $\theta \in \Theta = \{\theta|\sum_j \theta_j = 1, \theta_j \geq 0 \forall j\}$, denote $h_\theta = \sum_j \theta_j f_j$. For any convex combination $\theta$ over a set of candidate functions $\{h_1, \ldots, h_M\}$, we define the notation:

$$\ell_{\theta,g}(Y, Z, X) := \ell_{h_\theta,g}(Y, Z, X) \qquad\qquad R(\theta, g) := P\ell_{\theta,g}(Y, Z, X)$$

Here we define some optimal aggregates in the following sense:

$$j_\alpha^* := \underset{j=1,\ldots,M}{\arg\min} R(h_j, g_0) \qquad\qquad j^* := \underset{j=1,\ldots,M}{\arg\min} \|h_0 - h_j\|^2$$

$$\theta_\alpha^* := \underset{\theta \in \Theta}{\arg\min} R(h_\theta, g_0) \qquad\qquad \theta^* := \underset{\theta \in \Theta}{\arg\min} \|h_0 - h_\theta\|^2$$

$$h_\alpha^* := \arg\min R(h, g_0) \qquad\qquad h_{\alpha,\mathcal{H}}^* := \underset{h \in \mathcal{H}}{\arg\min} R(h, g_0)$$

*Proof of Theorem 7.*

$$\|h_{\hat{\theta}} - h_0\|^2 \leq 2\|h_{\hat{\theta}} - h_\alpha^*\|^2 + 2\|h_\alpha^* - h_0\|^2 \qquad\text{(By Strong Convexity)}$$

$$\leq \frac{2}{\alpha}\big(R(h_{\hat{\theta}}, g_0) - R(h_\alpha^*, g_0)\big) + O\big(\alpha^{\min\{2,\beta\}}\big)$$

$$= \frac{2}{\alpha}\big(R(h_{\hat{\theta}}, g_0) - R(h_{\alpha,\mathcal{H}_j}^*, g_0) + R(h_{\alpha,\mathcal{H}_j}^*, g_0) - R(h_\alpha^*, g_0)\big) + O\big(\alpha^{\min\{2,\beta\}}\big)$$

$$\text{(for any } j)$$

$$= \frac{2}{\alpha}\bigg(R(h_{\hat{\theta}}, g_0) - R(h_{j_\alpha^*}, g_0) + R(h_{j_\alpha^*}, g_0) - R(h_{\alpha,\mathcal{H}_j}^*, g_0) + R(h_{\alpha,\mathcal{H}_j}^*, g_0) - R(h_\alpha^*, g_0)\bigg)$$

$$+ O\big(\alpha^{\min\{2,\beta\}}\big)$$

$$\leq \frac{2}{\alpha}\bigg(R(h_{\hat{\theta}}, g_0) - R(h_{j_\alpha^*}, g_0) + R(h_j, g_0) - R(h_{\alpha,\mathcal{H}_j}^*, g_0) + R(h_{\alpha,\mathcal{H}_j}^*, g_0) - R(h_\alpha^*, g_0)\bigg)$$

$$+ O\big(\alpha^{\min\{2,\beta\}}\big)$$

When $\hat{g}$ is estimated using the standard MLE appraoch, we have that by Corollary 1 and Lemma 1, we have that:

$$R(h_j, g_0) - R(h^*_{\alpha, \mathcal{H}_j}, g_0) \leq c_1 \delta^2_{n,j} + c_2 (\delta^2_{n,j} + \epsilon_{\mathcal{G}})^{\frac{1}{2}} \|h_j - h^*_{\alpha, \mathcal{H}_j}\|$$

$$\leq c_1 \delta^2_{n,j} + \frac{c_2^2 (\delta^2_{n,j} + \epsilon_{\mathcal{G}})}{\alpha} + \frac{1}{2} \alpha \|h_j - h^*_{\alpha, \mathcal{H}_j}\|^2$$

$$\leq O\left( \delta^2_{n,j} + \frac{(\delta^2_{n,j} + \epsilon_{\mathcal{G}})}{\alpha} \right) \qquad \text{(By Eqn 17)}$$

Thus, we have $R(h_j, g_0) - R(h^*_{\alpha, \mathcal{H}_j}, g_0) \leq O\left( \frac{\delta^2_{n,j} + \epsilon_{\mathcal{G}}}{\alpha} \right)$. Instantiating this result for the function class $\mathcal{H}_M$, which denotes the convex hull when convex-ERM is used, or the set of candidate functions when best-ERM is used, we get that:

$$R(h_{\hat{\theta}}, g_0) - R(h_{j^*_\alpha}, g_0) \leq R(h_{\hat{\theta}}, g_0) - R(h_{\theta^*_\alpha}, g_0)$$

$$\leq \frac{\delta^2_{n,M} + \epsilon_{\mathcal{G}}}{\alpha}$$

where $\delta_{n,M} = \max\{\delta_{n,\mathcal{G}}, \delta_{n,\mathcal{H}_M}\}$. Since the function classes used to train the candidate functions are typically more complex than the convex hull over $M$ variables, it is safe to assume that $\delta_{n,\mathcal{H}_M} \leq \delta_{n,\mathcal{H}}$. Combining, we get:

$$\|h_{\hat{\theta}} - h_0\|^2 \leq O\left( \alpha^{\min\{2,\beta\}} + \frac{\delta^2_{n,j} + \epsilon_{\mathcal{G}}}{\alpha^2} \right) + \frac{2}{\alpha} \left( R(h^*_{\alpha, \mathcal{H}_j}, g_0) - R(h^*_\alpha, g_0) \right)$$

$$\leq O\left( \alpha^{\min\{2,\beta\}} + \frac{\delta^2_{n,j} + \epsilon_{\mathcal{G}}}{\alpha^2} \right) + \frac{2}{\alpha} \left( R(h, g_0) - R(h^*_\alpha, g_0) \right) \qquad \text{(for any } h \in \mathcal{H}_j)$$

For any function class $\mathcal{H}$, we have:

$$R(h, g_0) - R(h^*_\alpha, g_0) = \|\mathcal{T}(h - h^*_\alpha)\|^2 + \alpha \|h - h^*_\alpha\|^2$$

$$\leq 2\|\mathcal{T}(h - h_0)\|^2 + 2\|\mathcal{T}(h^*_\alpha - h_0)\|^2 + 2\alpha \|h - h_0\|^2 + 2\alpha \|h^*_\alpha - h_0\|^2$$

$$\leq 4\|h - h_0\|^2 + O\left( \|w_0\|^2 \alpha^{\min\{\beta+1,2\}} \right)$$

$$\text{(By Lemma 3 in Bennett et al. (2023b))}$$

Hence, for any function class $\mathcal{H}_j$, we can choose $h$ that attains $\min_{\mathcal{H}_j} \|h - h_0\| = \epsilon_{\mathcal{H}_j}$. Combining, we get that:

$$\|h_{\hat{\theta}} - h_0\|^2 \leq \min_j O\left( \alpha^{\min\{\beta+1,2\}-1} + \frac{\delta^2_{n,j} + \epsilon_{\mathcal{G}}}{\alpha^2} + \frac{1}{\alpha} \epsilon^2_{\mathcal{H}_j} \right). \qquad (\alpha \leq 1)$$

Analogously, if $\hat{g}$ is estimated using $\chi^2$-MLE, we have that by Corollary 2, Lemma 1 and Assumption 9:

$$R(h_j, g_0) - R(h^*_{\alpha, \mathcal{H}_j}, g_0) \leq O\left( \delta^2_n + (\delta^2_n + \epsilon_{\mathcal{G}})^{1/2} \|\hat{h} - h_*\|^\gamma_2 \right)$$

$$\leq O\left( \delta^2_n + \left( \frac{\delta^2_n + \epsilon_{\mathcal{G}}}{\alpha^\gamma} \right)^{\frac{1}{1-2\gamma}} + \alpha \|\hat{h} - h_*\|^2_2 \right)$$

$$\text{(By Young's Inequality)}$$

$$\leq O\left( \alpha \left( \frac{\delta^2_n + \epsilon_{\mathcal{G}}}{\alpha^{1-\gamma}} \right)^{\frac{1}{1-2\gamma}} + \alpha \left( \frac{\delta^2_n + \epsilon_{\mathcal{G}}}{\alpha^2} \right)^{1/(2-\gamma)} \right) \qquad \text{(By Eqn 18)}$$

$$\leq O\left( \alpha \left( \frac{\delta^2_n + \epsilon_{\mathcal{G}}}{\alpha^2} \right)^{1/(2-\gamma)} \right)$$

By the same argument for the standard MLE case, we get:

$$\|h_{\hat{\theta}} - h_0\|^2 \leq \min_j O\left(\alpha^{\min\{\beta+1,2\}-1} + \left(\frac{\delta_{n,M}^2 + \epsilon_{\mathcal{G}}}{\alpha^2}\right)^{1/(2-\gamma)} + \frac{1}{\alpha}\epsilon_{\mathcal{H}_j}^2\right)$$

$\square$

## G    PROOF OF THEOREM 8 AND 12

In this section, we prove the convergence rate of iterative RMIV in Section 8 under a unified framework. We prove the results of Theorem 12 and 12 respectively. Recall that we define

$$h_{m,*} = \arg\min_{h \in \mathcal{H}} \mathbb{E}[Y - \mathcal{T}h(Z)^2] + \alpha \cdot \mathbb{E}[(h - h_{m-1,*})^2(X)],$$

by Lemma 6 and Assumption 2, we have

$$\|h_{m,*} - h_0\|_2^2 \leq \|w_0\|_2^2 \alpha^{\min\{\beta,2m\}}.$$

Therefore, we only need to provide a upper bound for $\|\hat{h}_m - h_{m,*}\|_2^2$, and then choose the proper $\alpha$ deliberately. We start by proving the following lemma, and with the different convergence rate of MLE and $\chi^2$-MLE, we conclude the proof of Theorem 8 and Theorem 12 respectively.

**Lemma 3.** *We have the following inequality holds with probability at least* $1 - m\exp(n\delta_{n,\mathcal{H}}^2)$*:*

$$\|\hat{h}_m - h_{m,*}\|^2 \leq O(\delta_{n,\mathcal{H}}^2/\alpha^2) + O\left(\frac{\mathbb{E}[|(\mathcal{T} - \hat{\mathcal{T}})(\hat{h}_m - h_{m,*})|]}{\alpha}\right) + 16\|\hat{h}_{m-1} - h_{m-1,*}\|^2.$$

*Proof.* Recall that our solution $\hat{h}_m$ satisfies

$$\hat{h}_m = \arg\min_{h \in \mathcal{H}} L(\hat{\mathcal{T}}h) + \alpha\mathbb{E}_n[\{h - \hat{h}_{m-1}\}^2].$$

We define

$$L_m(\tau) = \mathbb{E}[\mathbb{E}[h_0 - h_{m,*} - \tau(\hat{h}_m - h_{m,*}) \mid Z]^2] + \alpha\|h_{m,*} + \tau(\hat{h}_m - h_{m,*}) - h_{m-1,*}\|^2,$$

By definition, $L_m(\tau)$ is minimized by $\tau = 0$. Note that by strong convexity and property of quadratic function, we have

$$L_m(1) - L_m(0) = L'(0) + L''(0) \geq L''(0),$$

Therefore

$$\alpha\|\hat{h}_m - h_{m,*}\|^2 + \|\mathcal{T}(\hat{h}_m - h_{m,*})\|^2$$
$$\leq \|\mathcal{T}(h_0 - \hat{h}_m)\|^2 - \|\mathcal{T}(h_0 - h_{m,*})\|^2 + \alpha(\|\hat{h}_m - h_{m-1,*}\|^2 - \|h_{m,*} - h_{m-1,*}\|^2)$$
$$= \mathbb{E}[L(\mathcal{T}\hat{h}_m)] - \mathbb{E}[L(\mathcal{T}h_{m,*})] + \alpha(\|\hat{h}_m - h_{m-1,*}\|^2 - \|h_{m,*} - h_{m-1,*}\|^2),$$

and thus we have

$$\alpha\|\hat{h}_m - h_{m,*}\|^2 + \|\mathcal{T}(\hat{h}_m - h_{m,*})\|^2$$
$$\leq \mathbb{E}[L(\hat{\mathcal{T}}\hat{h}_m)] - \mathbb{E}[L(\hat{\mathcal{T}}h_{m,*})] + c \cdot \mathbb{E}[|(\mathcal{T} - \hat{\mathcal{T}})(\hat{h}_m - h_{m,*})|]$$
$$\quad + \alpha(\|\hat{h}_m - h_{m-1,*}\|^2 - \|h_{m,*} - h_{m-1,*}\|^2)$$
$$\leq |(\mathbb{E} - \mathbb{E}_n)(L(\hat{\mathcal{T}}\hat{h}_m) - L(\hat{\mathcal{T}}h_{m,*}))| + \mathbb{E}_n[L(\hat{\mathcal{T}}\hat{h}_m) - L(\hat{\mathcal{T}}h_{m,*})]$$
$$\quad + c \cdot \mathbb{E}[|(\mathcal{T} - \hat{\mathcal{T}})(\hat{h}_m - h_{m,*})|] + \alpha(\|\hat{h}_m - h_{m-1,*}\|^2 - \|h_{m,*} - h_{m-1,*}\|^2)$$
$$\leq c_1(\delta_n\|\hat{\mathcal{T}}(\hat{h}_m - h_{m,*})\| + \delta_n^2) + \mathbb{E}_n[L(\hat{\mathcal{T}}\hat{h}_m) - L(\hat{\mathcal{T}}h_{m,*})] + c \cdot \mathbb{E}[|(\mathcal{T} - \hat{\mathcal{T}})(\hat{h}_m - h_{m,*})|]$$
$$\quad + \alpha(\|\hat{h}_m - h_{m-1,*}\|^2 - \|h_{m,*} - h_{m-1,*}\|^2),$$

holds for all $m$ simultaneously with probability at least $1 - m\exp(n\delta_{n,\mathcal{H}}^2)$, recall that $\delta_{n,\mathcal{H}}^2$ is the critical radius. Here the second inequality comes from triangular inequality and $L(\cdot)$ being $O(1)$-Lipschitz, the third inequality comes from Lemma 4. By Eq. equation 10,

$$\mathbb{E}_n[L(\hat{\mathcal{T}}\hat{h}_m) - L(\hat{\mathcal{T}}h_{m,*})] \leq \alpha(\|h_{m,*} - \hat{h}_{m-1}\|_n^2 - \|\hat{h}_m - \hat{h}_{m-1}\|_n^2),$$

therefore we have

$$\alpha\|\hat{h}_m - h_{m,*}\|^2 + \|\mathcal{T}(\hat{h}_m - h_{m,*})\|^2$$

$$\leq c_1(\delta_n\|(\hat{h}_m - h_{m,*})\| + \delta_n^2) + \alpha(\|h_{m,*} - \hat{h}_{m-1}\|_n^2 - \|\hat{h}_m - \hat{h}_{m-1}\|_n^2) + c \cdot \mathbb{E}[|(\mathcal{T} - \hat{\mathcal{T}})(\hat{h}_m - h_{m,*})|]$$

$$+ \alpha\big(\|\hat{h}_m - h_{m-1,*}\|^2 - \|h_{m,*} - h_{m-1,*}\|^2\big).$$

We are now interested in bounding

$$(\|h_{m,*} - \hat{h}_{m-1}\|_n^2 - \|\hat{h}_m - \hat{h}_{m-1}\|_n^2) + \big(\|\hat{h}_m - h_{m-1,*}\|^2 - \|h_{m,*} - h_{m-1,*}\|^2\big).$$

We divide it into two terms:

$$I_1 := \big(\|h_{m,*} - \hat{h}_{m-1}\|^2 - \|\hat{h}_m - \hat{h}_{m-1}\|^2\big) + \big(\|\hat{h}_m - h_{m-1,*}\|^2 - \|h_{m,*} - h_{m-1,*}\|^2\big),$$

$$I_2 := (\|h_{m,*} - \hat{h}_{m-1}\|_n^2 - \|\hat{h}_m - \hat{h}_{m-1}\|_n^2) - (\|h_{m,*} - \hat{h}_{m-1}\|^2 - \|\hat{h}_m - \hat{h}_{m-1}\|^2)$$

Note that $|I_1| = \big|2\langle\hat{h}_{m-1} - h_{m-1,*}, \hat{h}_m - h_{m,*}\rangle\big|$, we have

$$I_1 \leq 2\|\hat{h}_{m-1} - h_{m-1,*}\|_2\|\hat{h}_m - h_{m,*}\|_2,$$

For $I_2$, we divide it into two terms $I_3$ and $I_4$, defined by

$$I_3 := \|h_{m,*} - \hat{h}_{m-1}\|_n^2 - \|h_{m,*} - h_{m-1,*}\|_n^2 - (\|h_{m,*} - \hat{h}_{m-1}\|^2 - \|h_{m,*} - h_{m-1,*}\|^2),$$

$$I_4 := \|h_{m,*} - h_{m-1,*}\|_n^2 - \|\hat{h}_m - \hat{h}_{m-1}\|_n^2 - (\|h_{m,*} - h_{m-1,*}\|^2 - \|\hat{h}_m - \hat{h}_{m-1}\|^2)$$

Since each of these is the difference of two centered empirical processes, that are also Lipschitz losses (since $h_{m,*}, \hat{h}_m, h_{m-1,*}, \hat{h}_{m-1}$ are uniformly bounded) and since $h_{m,*}$ is a population quantity and not dependent on the empirical sample that is used for the $m$-th iterate, we can also upper bound these,

$$I_3 = O(\delta_{n,\mathcal{H}}^2\|\hat{h}_{m-1} - h_{m-1,*}\| + \delta_{n,\mathcal{H}}^2),$$

$$I_4 = O(\delta_{n,\mathcal{H}}\|\hat{h} - h_{m,*} + h_{m-1,*} - \hat{h}_{m-1}\| + \delta_{n,\mathcal{H}}^2) = O\left(\delta_{n,\mathcal{H}}(\|\hat{h} - h_{m,*}\| + \|h_{m-1,*} - \hat{h}_{m-1}\| + \delta_{n,\mathcal{H}}^2)\right),$$

combine everything together, we can prove that

$$(\|h_{m,*} - \hat{h}_{m-1}\|_n^2 - \|\hat{h}_m - \hat{h}_{m-1}\|_n^2) + \big(\|\hat{h}_m - h_{m-1,*}\|^2 - \|h_{m,*} - h_{m-1,*}\|^2\big)$$

$$\leq O(\delta_n^2 + \delta_n(\|\hat{h}_m - h_{m,*}\| + \|\hat{h}_{m-1} - h_{m-1,*}\|)) + 2\|\hat{h}_{m-1} - h_{m-1,*}\|\|\hat{h}_m - h_{m,*}\|.$$

Therefore, we have

$$\alpha\|\hat{h}_m - h_{m,*}\|^2$$

$$\leq O\left(\delta_{n,\mathcal{H}}^2 + \delta_{n,\mathcal{H}}\|\hat{h}_m - h_{m,*}\| + c \cdot \mathbb{E}[|(\mathcal{T} - \hat{\mathcal{T}})(\hat{h}_m - h_{m,*})|] + \alpha\delta_{n,\mathcal{H}}(\|\hat{h}_m - h_{m,*}\| + \|\hat{h}_{m-1} - h_{m-1,*}\|))\right)$$

$$+ 2\alpha\|\hat{h}_{m-1} - h_{m-1,*}\|\|\hat{h}_m - h_{m,*}\|.$$

By applying AM-GM inequality and utilizing $\alpha \leq 1$, we have

$$\frac{\alpha}{8}\|\hat{h}_m - h_{m,*}\|^2 \leq O\big(\delta_n^2/\alpha + \delta_n^2 + \alpha\delta_n^2\big) + c \cdot \mathbb{E}[|(\mathcal{T} - \hat{\mathcal{T}})(\hat{h}_m - h_{m,*})|] + 2\alpha\|\hat{h}_{m-1} - h_{m-1,*}\|^2,$$

therefore we have

$$\|\hat{h}_m - h_{m,*}\|^2 \leq O\big(\delta_n^2/\alpha^2 + \delta_n^2/\alpha\big) + O\left(\frac{\mathbb{E}[|(\mathcal{T} - \hat{\mathcal{T}})(\hat{h}_m - h_{m,*})|]}{\alpha}\right) + 16\|\hat{h}_{m-1} - h_{m-1,*}\|^2.$$

**Proof for Theorem 8.** By Corollary 1, we have

$$\mathbb{E}[|(\mathcal{T} - \hat{\mathcal{T}})(\hat{h}_m - h_{m,*})|] = \|(\mathcal{T} - \hat{\mathcal{T}})(\hat{h}_m - h_{m,*})\|_1 \le \delta_n \cdot \|\hat{h}_m - h_{m,*}\|_2,$$

therefore by Lemma 3, we have

$$\|\hat{h}_m - h_{m,*}\|^2 \le O(\delta_n^2/\alpha^2 + \delta_n\|\hat{h}_m - h_{m,*})\|_2) + 16\|\hat{h}_{m-1} - h_{m-1,*}\|^2.$$

By Lemma 7, we have

$$\|\hat{h}_m - h_{m,*}\|^2 \le 4O(\delta_n^2/\alpha^2) + 16\|\hat{h}_{m-1} - h_{m-1,*}\|^2$$
$$\le 128^m \cdot \delta_n^2/\alpha^2,$$

where the second inequality comes from induction. Therefore, by Lemma 6, we have

$$\|\hat{h}_m - h_0\|^2 = O(128^m \cdot \delta_n^2/\alpha^2 + \alpha^{\min\{\beta, 2m\}}).$$

Set $\alpha = \delta_n^{\frac{2}{2+\min\{\beta, 2m\}}}$, and we conclude the proof.

**Proof for Theorem 12** By Assumption 9, we have $\|\hat{h}_m - h_{m,*}\|_\infty \le \|\hat{h}_m - h_{m,*}\|_2^\gamma$, which implies

$$\|\hat{h}_m - h_{m,*}\|^2 \le O(\delta_n^2/\alpha^2 + \delta_n^2/\alpha) + O\left(\delta_n/\alpha \cdot \|\hat{h}_m - h_{m,*}\|^\gamma\right) + 16\|\hat{h}_{m-1} - h_{m-1,*}\|^2,$$

by Lemma 7, we have

$$\|\hat{h}_m - h_{m,*}\|^2 \le 4\max\left\{O(\delta_n^2/\alpha^2 + 16\|\hat{h}_{m-1} - h_{m-1,*}\|^2), O((\delta_n/\alpha)^{2/(2-\gamma)})\right\}$$
$$\le O(128^m \max\left\{\delta_n^2/\alpha^2, (\delta_n/\alpha)^{2/(2-\gamma)}\right\}),$$

where the second inequality comes from induction. Therefore, by Lemma 6, we have

$$\|\hat{h}_m - h_0\|^2 = O(128^m \cdot \max\left\{\delta_n^2/\alpha^2, (\delta_n/\alpha)^{2/(2-\gamma)}\right\} + \alpha^{\min\{\beta, 2m\}}).$$

Set $\alpha = \delta_n^{\frac{2}{2+(2-\gamma)\min\{\beta, 2m\}}}$, Then $\delta_n/\alpha = O(\delta_n^{\frac{(2-\gamma)\min\{\beta, 2m\}}{2+(2-\gamma)\min\{\beta, 2m\}}}) \lesssim 1$, and since $\gamma \in (0, 1)$, we have

$$\max\left\{\delta_n^2/\alpha^2, (\delta_n/\alpha)^{2/(2-\gamma)}\right\} = (\delta_n/\alpha)^{2/(2-\gamma)},$$

and

$$\|\hat{h}_m - h_0\|^2 = O(128^m \cdot \delta_n^{\frac{2\min\{\beta, 2m\}}{2+(2-\gamma)\min\{\beta, 2m\}}}),$$

and we conclude the proof of Theorem 12. $\qquad\square$

## H CONVERGENCE RATE OF MLE AND $\chi^2$-MLE

### H.1 CONVERGENCE RATE OF MLE

In this section, we aim to characterize the convergence rate of conditional MLE equation 4 in terms of the critical radius $\delta_{n,\mathcal{G}}$ of function class $\mathcal{G}$ and model misspecification. Specifically, we prove the following Theorem:

**Theorem 13** (Convergence rate for misspecified MLE). *Suppose Assumption 4 and condition in Theorem 5 holds, and there exists $g^\dagger \in \mathcal{G}$ such that $\mathbb{E}_{z \sim g_0}[D_{KL}(g_0(\cdot|z), g^\dagger(\cdot|z))] \le \epsilon_{\mathcal{G}}$. Then we have*

$$\mathbb{E}_{z \sim g(z)}\left[H^2(\hat{g}(\cdot|z)|g_0(\cdot|z))\right] \le \delta_n^2 + \epsilon_{\mathcal{G}}$$

*holds with probability at least $1 - c_1 \exp(c_2 \frac{c_0}{C+c_0} n\delta_n^2)$.*

*Proof.* We work with the transformed function class $\mathcal{F} = \left\{\sqrt{\frac{g+g_0}{2g_0}} \middle| g \in \mathcal{G}\right\}$, and define $\mathcal{L}_f = -\log f(x)$ for $f \in \mathcal{F}$. Note that $\mathcal{F}$ is a function class whose element maps $\mathcal{X} \times \mathcal{Z}$ to $\mathbb{R}$. We define the population version of localized Rademacher complexity for function class $\mathcal{F}^* := \text{star}((\mathcal{F} -$

$f^*) \cup \{0\}$). By Assumption 4 and 1-boundedness of $\mathcal{G}$, $\mathcal{F}$ and $\mathcal{F}^*$ are bounded by a constant $b := \frac{C_0 + C}{2C_0}$ in $\|\cdot\|_\infty$. The critical radius $\delta_{n,\mathcal{F}}$ of function class $\mathcal{F}^*$ is any solution such that

$$\delta^2 \geq c/n \text{ and } \bar{R}_n(\delta; \mathcal{F}^*) \leq \delta^2/b.$$

Such critical radius can be easily calculated for a large number of function classes. For example, we can use

$$\frac{64}{\sqrt{n}} \int_{\delta^2/2b}^{\delta} \sqrt{\log N_n(t, \mathcal{B}(\delta, \mathcal{F}^*))} dt \leq \frac{\delta^2}{b}$$

to calculate $\delta_{n,\mathcal{F}}$, where $\mathcal{B}(\delta, \mathcal{F}^*) := \{f \in \mathcal{F}^* \mid \|f\|_2 \leq \delta\}$, $N_n$ is the empirical covering number conditioned on $\{(x_i, z_i)\}_{i \in [n]}$. For a cost function $\mathcal{L} : \mathbb{R} \to \mathbb{R}$, we define $\mathcal{L}_f(x,z) := \mathcal{L}(f(x,z))$. We make the following definition.

**Definition 1.** *We say $\mathcal{L}_f$ is $\gamma$-strongly convexity at $f^*$ if*

$$\mathbb{E}_{z \sim g_0(z), x \sim g_0(x|z)}\left[\mathcal{L}_f(x,z) - \mathcal{L}_{f^*}(x,z) - \partial\mathcal{L}_{f^*}(x,z)(f - f^*)(x,z)\right] \geq \frac{\gamma}{2}\|f - f^*\|_2^2$$

*for all $f \in \mathcal{F}$.*

Note that for any $f \in \mathcal{F}$ we have and $|\log f(x) - \log f'(x)| \leq \sqrt{2}|f(x) - f'(x)|$ since $\|f\|_\infty \geq 1/\sqrt{2}$. By the definition of Hellinger distance, we have

$$\|f - f^*\|_2^2 = \mathbb{E}_{z \sim g_0(z)}\left[H^2\left(\frac{g + g_0}{2}\Big| g_0\right)\right],$$

and since $H^2(g_1 \mid g_2) \leq 2D_{\mathrm{KL}}(f_1 \mid f_2)$, we have $\|f - f^*\|_2^2 \leq \mathbb{P}(\mathcal{L}_f - \mathcal{L}_{f^*})$, thus $\mathcal{L}$ is 2-strongly convex at $f^*$. Utilizing strong convexity and Lemma 4, we have the following inequality holds with probability $1 - \exp(n\delta_{n,\mathcal{F}}^2)$:

$$\|\hat{f} - f_0\|_2^2 \leq 2\mathbb{E}_{z \sim g_0(z), x \sim g_0(x|z)}[\mathcal{L}_{\hat{f}}(x,z) - \mathcal{L}_{f_0}(x,z)]$$

$$= 2\mathbb{E}_{z \sim g_0(z), x \sim g_0(x|z)}[\mathcal{L}_{\hat{f}}(x,z) - \mathcal{L}_{f^\dagger}(x,z)] + 2\mathbb{E}_{z \sim g_0(z),}[D_{\mathrm{KL}}(g_0(\cdot|z) \mid (g^\dagger + g_0)/2(\cdot|z))]$$

$$\leq 2(\mathbb{E}_n - \mathbb{E})[\mathcal{L}_{\hat{f}}(x,z) - \mathcal{L}_{f^\dagger}(x,z)] + 2\mathbb{E}_n[\mathcal{L}_{\hat{f}}(x,z) - \mathcal{L}_{f^\dagger}(x,z)]$$

$$+ \mathbb{E}_{z \sim g_0(z)}[D_{\mathrm{KL}}(g_0(\cdot|z) \mid g^\dagger(\cdot|z))]$$

$$\leq O(\delta_{n,\mathcal{F}}\|\hat{f} - f^\dagger\|_2 + \delta_{n,\mathcal{F}}^2) + \mathbb{E}_{z \sim g_0(z)}[D_{\mathrm{KL}}(g_0(\cdot|z) \mid g^\dagger(\cdot|z))]$$

$$\leq O(\delta_{n,\mathcal{F}}\|\hat{f} - f_0\|_2 + \delta_{n,\mathcal{F}}\|f_0 - f^\dagger\|_2 + \delta_{n,\mathcal{F}}^2) + \mathbb{E}_{z \sim g_0(z)}[D_{\mathrm{KL}}(g_0(\cdot|z) \mid g^\dagger(\cdot|z))],$$

here the first inequality comes from strong convexity, the third inequality comes from $\log(\frac{2x}{x+y}) \leq \frac{1}{2}\log(\frac{x}{y})$ and the definition of MLE. The forth inequality comes from Lemma 4. Solve this inequality, and recall that $\|f - h_0\|_2^2 = \mathbb{E}_{z \sim g_0(z)}[H^2((g + g_0)(\cdot|z)/2 \mid g_0(\cdot|z))]$, we have

$$\mathbb{E}_{z \sim g_0(z)}[H^2(\hat{g}(\cdot|z) \mid g_0(\cdot|z))] \leq O(\delta_{n,\mathcal{F}}^2 + \delta_{n,\mathcal{F}}\|f_0 - f^\dagger\|_2 + \mathbb{E}_{z \sim g_0(z)}[D_{\mathrm{KL}}(g_0(\cdot|z) \mid g^\dagger(\cdot|z))])$$

$$\leq O(\delta_{n,\mathcal{F}}^2 + \delta_{n,\mathcal{F}}\mathbb{E}_{z \sim g_0(z)}[D_{\mathrm{KL}}(g_0(\cdot|z), g^\dagger(\cdot|z))]^{1/2}$$

$$+ \mathbb{E}_{z \sim g_0(z)}[D_{\mathrm{KL}}(g_0(\cdot|z) \mid g^\dagger(\cdot|z))])$$

$$\leq O(\delta_{n,\mathcal{F}}^2 + \mathbb{E}_{z \sim g_0(z)}[D_{\mathrm{KL}}(g_0(\cdot|z) \mid g^\dagger(\cdot|z))]),$$

here the first inequality comes from Lemma 5, the second inequality comes from Lemma 9. Thus we conclude the proof of Theorem 13. □

We provide the following corollary, which would help characterize the $L_1$ and $L_2$ error of $\mathcal{T}h$ introduced by MLE.

**Corollary 1.** *Under Assumption 4, for all $h' \in \mathcal{H} - \mathcal{H}$, we have $\|(\hat{\mathcal{T}} - \mathcal{T})h'\|_1 \leq \{1/c_0 + 1\}\|h'\|_2 \cdot (\delta_{n,\mathcal{H}}^2 + \epsilon_\mathcal{G})^{1/2}$ and $\|(\hat{\mathcal{T}} - \mathcal{T})h'\|_2 \leq (C_{2,4}C)^{1/2} \cdot (C/c_0 + 1)\|h'\|_2 \cdot (\delta_{n,\mathcal{G}}^2 + \epsilon_\mathcal{G})^{1/4}$ with probability at least $1 - c_2\exp(c_3 n\delta_{n,\mathcal{G}}^2)$.*

*Proof.* We first prove the bound for $L_1$ error $\|(\hat{\mathcal{T}} - \mathcal{T})h'\|_1$. We have the following inequality:

$$\|(\hat{\mathcal{T}} - \mathcal{T})h'\|_1 = \mathbb{E}_{z \sim g_0(z)} \left[ |\mathbb{E}_{x \sim g_0(x|z)} \left[ \frac{\hat{g}(x|z)}{g_0(x|z)} h'(x) - h'(x) \right] | \right]$$

$$\leq \mathbb{E}_{z \sim g_0(z), x \sim g_0(x|z)} \left[ |\frac{\hat{g}(x|z)}{g_0(x|z)} h'(x) - h'(x)| \right]$$

$$\leq \mathbb{E}_{z \sim g_0(z), x \sim g_0(x|z)} \left[ \sqrt{\frac{\hat{g}(x|z)}{g_0(x|z)}} |h'(x)| |\sqrt{\frac{\hat{g}(x|z)}{g_0(x|z)}} - 1| \right]$$

$$+ \mathbb{E}_{z \sim g_0(z), x \sim g_0(x|z)} \left[ |h'(x)| |\sqrt{\frac{\hat{g}(x|z)}{g_0(x|z)}} - 1| \right]$$

$$\leq \mathbb{E}[\frac{\hat{g}(x|z)}{g_0(x|z)} h'^2(x)]^{1/2} \times \mathbb{E}[2H^2(\hat{g}(\cdot|z) \mid g_0(\cdot|z))]$$

$$+ \mathbb{E}[\frac{\hat{g}(x|z)}{g(^\star(x|z)} h'^2(x)]^{1/2} \cdot \mathbb{E}[2H^2(\hat{g}(\cdot|z) \mid g_0(\cdot|z))]^{1/2} \qquad \text{(CS inequality)}$$

$$\leq 2\{1/c_0 + 1\}\mathbb{E}[h^2(x)]^{1/2} \cdot \mathbb{E}[2H^2(\hat{g}(\cdot|z) \mid g_0(\cdot|z))]^{1/2}$$

$$= \{1/c_0 + 1\}\|h'\|_2 \cdot (\delta_{n,\mathcal{G}}^2 + \epsilon_{\mathcal{G}})^{1/2}.$$

where the second inequality comes from Assumption 4. Next, we prove the upper bound for $L_2$ error $\|(\hat{\mathcal{T}} - \mathcal{T})h'\|_2$. We have

$$\|(\hat{\mathcal{T}} - \mathcal{T})h'\|_2 = \left\{ \mathbb{E}[|(\mathcal{T} - \hat{\mathcal{T}})h'|^2] \right\}^{1/2}$$

$$\leq 2C_Y \|(\mathcal{T} - \hat{\mathcal{T}})h'\|_1^{1/2}$$

$$\leq 2C_Y \delta_{n,\mathcal{H}}^{1/2} \|h'\|^{1/2}.$$

and we conclude the proof. $\qquad \square$

## H.2 Convergence rate of $\chi^2$-MLE

For the convergence rate of $\chi^2$-MLE, we present the following theorem:

**Theorem 14** (Convergence rate for $\chi^2$-MLE, Corollary 14.24 of Wainwright (2019) ). *For $\hat{g}$ generated by 11, we have*

$$\mathbb{E}_{z \sim g_0(z)} \left[ \{ \int |\hat{g}(x|z) - g_0(x|z)| d\mu(x)\}^2 \right] = O\left( \delta_{n,\mathcal{G}}^2 + \inf_{g \in \mathcal{G}} \mathbb{E}_{z \sim g_0(z)} \left[ \{ \int |g(x|z) - g_0(x|z)| d\mu(x)\}^2 \right] \right)$$

*with probability at least $1 - c_1 \exp(c_2 n \delta_{n,\mathcal{G}}^2)$.*

*Proof.* By Theorem 13.13 of Wainwright (2019), we have

$$\mathbb{E}_n \left[ \{ \int |\hat{g}(x|z) - g_0(x|z)| d\mu(x)\}^2 \right] = O\left( \delta_{n,\mathcal{G}}^2 + \inf \mathbb{E}_n \left[ \{ \int |g(x|z) - g_0(x|z)| d\mu(x)\}^2 \right] \right)$$

holds with probability at least $1 - \exp(c_1 n \delta_{n,\mathcal{G}}^2)$. By Theorem 15, we have

$$(\mathbb{E}_n - \mathbb{E}) \left[ \{ \int |g(x|z) - g_0(x|z)| d\mu(x)\}^2 \right] \leq O(\delta_{n,\mathcal{F}}^2)$$

holds for all $g \in \mathcal{G}$ with probability at least $1 - c_2 \exp(c_3 n \delta_{n,\mathcal{G}}^2)$, and the proof is done. and the proof is done. $\qquad \square$

We provide the following corollary, which would help characterize the error introduced by $\chi^2$-MLE.

**Corollary 2.** *With $\chi^2$-MLE, we have the following inequality holds for all $h \in \mathcal{H}$ with probability at least $1 - c_2 \exp(c_3 n \delta_{n,\mathcal{G}}^2)$:*

$$\|(\mathcal{T} - \hat{\mathcal{T}})h\|_2^2 \leq (\delta_{n,\mathcal{G}}^2 + \epsilon_{\mathcal{G}})\|h\|_\infty^2.$$

*Proof.* By

$$\|(\mathcal{T} - \hat{\mathcal{T}})h\|_2^2 = \mathbb{E}_{z \sim g_0(z)}\left[\left(\int_{\mathcal{X}}\{\hat{g}(x|z) - g_0(x|z)\}h(x)d\mu(x)\right)^2\right]$$
$$\leq (\delta_{n,\mathcal{G}}^2 + \epsilon_{\mathcal{G}})\|h\|_\infty^2.$$

We conclude the proof. □

## I    AUXILIARY LEMMA

We introduce the following lemma, which gives a uniform convergence rate of loss error.

**Lemma 4** (Localized Concentration, Foster and Syrgkanis (2019))**.** *For any $f \in \mathcal{F} := \times_{i=1}^d \mathcal{F}_i$ be a multivalued outcome function, that is almost surely absolutely bounded by a constant. Let $\ell(Z; f(X)) \in \mathbb{R}$ be a loss function that is $O(1)$-Lipschitz in $f(X)$, with respect to the $\ell_2$ norm. Let $\delta_n = \Omega\left(\sqrt{\frac{d \log\log(n) + \log(1/\zeta)}{n}}\right)$ be an upper bound on the critical radius of $\mathrm{star}(\mathcal{F}_i)$ for $i \in [d]$. Then for any fixed $h_0 \in \mathcal{F}$, w.p. $1 - \zeta$ :*

$$\forall f \in \mathcal{F}: |(\mathbb{E}_n - \mathbb{E})\left[\ell(Z; f(X)) - \ell(Z; h_0(X))\right]| = O\left(d\delta_n\sum_{i=1}^d \|f_i - f_{i,0}\|_2 + d\delta_n^2\right)$$

*If the loss is linear in $f(X)$, i.e. $\ell(Z; f(X) + f'(X)) = \ell(Z; f(X)) + \ell(Z; f'(X))$ and $\ell(Z; \alpha f(X)) = \alpha\ell(Z; f(X))$ for any scalar $\alpha$, then it suffices that we take $\delta_n = \Omega\left(\sqrt{\frac{\log(1/\zeta)}{n}}\right)$ that upper bounds the critical radius of $\mathrm{star}(\mathcal{F}_i)$ for $i \in [d]$.*

*Proof.* For a detailed proof, please refer to Foster and Syrgkanis (2019). □

The following lemma is useful when proving the convergence rate of Hellinger distance.

**Lemma 5** (Lemma 4.1 in Van de Geer (1993))**.** *For two density functions $g_1$ and $g_2$, define $g_u = ug_1 + (1-u)g_2$, then we have*

$$\frac{1}{4(1-u)}H^2(g_1 \mid g_u) \leq H^2(g_1 \mid g_2) \leq \frac{1}{(1-u)^2}H^2(g_1 \mid g_u)$$

*holds for all $u \in (0,1)$*

*Proof.* For a detailed proof, see Lemma 4.1 in Van de Geer (1993). □

**Lemma 6** (Lemma 5 in Bennett et al. (2023b))**.** *If $h_0$ is the minimum $L_2$-norm solution to the linear inverse problem and satisfies the $\beta$-source condition, then the solution to the $t$-th iterate of Tikhonov regularization $h_{m,*}$, defined in Equation equation 9, with $h_{0,*} = 0$, satisfies that*

$$\|h_{m,*} - h_0\|^2 \leq \|w_0\|^2\alpha^{\min\{\beta,2t\}}, \qquad \|\mathcal{T}h_{m,*} - \mathcal{T}h_0\|^2 \leq \|w_0\|^2\alpha^{\min\{\beta+1,2t\}}.$$

*Proof.* For a detailed proof, see Lemma 5 in Bennett et al. (2023b). □

The following lemma upper-bounds the bias introduced by Tikhonov regularization.

**Lemma 7.** *For*
$$x^2 \leq c_1 + c_2 x^{\gamma_1} + c_3 x^{\gamma_2},$$
*where $c_1, c_2 > 0$, $0 \leq \gamma \leq 1$, we have $x \leq 3\max\left\{\sqrt{c_1}, c_2^{1/(2-\gamma_1)}, c_3^{1/(2-\gamma_2)}\right\}$.*

*Proof.* Since $x^2 - c_2 x^{\gamma_1} - c_3 x^{\gamma_2} - c_1$ is a convex function with negative intercept, we only need to prove that for $x_0 = 3\max\left\{\sqrt{c_1}, c_2^{1/(2-\gamma_1)}, c_3^{1/(2-\gamma_2)}\right\}$, we have $x_0^2 - c_2 x_0^{\gamma_1} - c_3 x_0^{\gamma_2} - c_1 \geq 0$. For simplicity, we consider $\sqrt{c_1} \geq \max\{c_2^{1/(2-\gamma_1)}, c_3^{1/(2-\gamma_2)}\}$, and we have

$$x_0^2 = 9c_1 \geq c_1 + c_2 \cdot 3^{\gamma_1}c_1^{\gamma_1/2} + c_3 \cdot 3^{\gamma_2}c_1^{\gamma_2/2} = c_1 + c_2 x_0^{\gamma_1} + c_3 x_0^{\gamma_2},$$

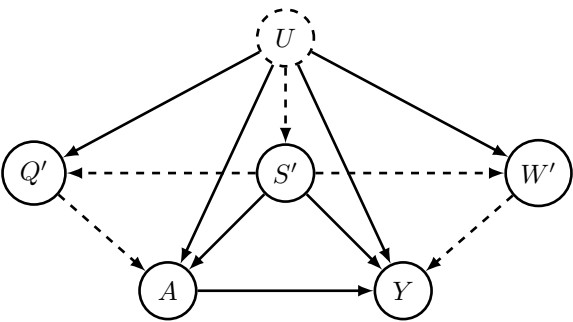

Figure 1: A typical causal diagram for negative controls. The dashed edges may be absent, and the dashed circle around $S'$ indicates that $U$ is unobserved.

similarly we have the same result when $c_2^{1/(2-\gamma_1)} \geq \max\{\sqrt{c_1}, c_3^{1/(2-\gamma_2)}\}$ or $c_3^{1/(2-\gamma_2)} \geq \max\{\sqrt{c_1}, c_2^{1/(2-\gamma_1)}\}$, and we conclude the proof. $\qquad\square$

Next, we introduce the following lemma that gives a uniform convergence rate for function class $\mathcal{F}$, which is adapted from Wainwright (2019).

**Lemma 8** (Theorem 14.20 in Wainwright (2019).)**.** *Suppose we have a 1-uniformly bounded function class $\mathcal{F}$ that is star-shaped around a population minimizer $f^*$. Let $\delta_n \geq \frac{c}{n}$ be the solution to the inequality*

$$\bar{R}_n(\delta; \mathcal{F}^*) \leq \delta^2.$$

*Suppose the loss function $\mathcal{L}_f$ is L-Lipschitz, then with probability at least $1 - c_1 \exp(-c_2 n \delta_{n,\mathcal{F}}^2/b)$, either of the following events holds for all $f \in \mathcal{F}$:*

*(1) $\|f - f^*\|_2 \leq \delta_n$;*

*(2) $|\mathbb{P}_n(\mathcal{L}_f - \mathcal{L}_{f^*}) - \mathbb{P}(\mathcal{L}_f - \mathcal{L}_{f^*})| \leq 10 L \delta_n \|f - f^*\|_2$.*

The following lemma is a classical result for localization and uniform laws.

**Theorem 15** (Theorem 14.1 of Wainwright (2019).)**.** *Given a star-shaped and b-uniformly bounded function class $\mathcal{F}$, let $\delta_n$ be any positive solution of the inequality*

$$\bar{\mathcal{R}}_n(\delta; \mathcal{F}) \leq \frac{\delta^2}{b}.$$

*Then for any $t \geq \delta_n$, we have*

$$\left| \|f\|_n^2 - \|f\|_2^2 \right| \leq \frac{1}{2} \|f\|_2^2 + \frac{t^2}{2} \quad \text{for all } f \in \mathcal{F}$$

*with probability at least $1 - c_1 e^{-c_2 \frac{n\delta_n^2}{b^2}}$. If in addition $n\delta_n^2 \geq \frac{2}{c_2} \log(4 \log(1/\delta_n))$, then*

$$\left| \|f\|_n - \|f\|_2 \right| \leq c_0 \delta_n \quad \text{for all } f \in \mathcal{F}$$

*with probability at least $1 - c_1' e^{-c_2' \frac{n_0^2}{b^2}}$.*

The next lemma enables us to upper-bound KL divergence by Hellinger distance.

**Lemma 9** (Example 14.10 in Wainwright (2019). )**.** *For any two density function $g_1$ and $g_2$, we have*

$$H^2(g_1 \mid g_2) \leq 2 D_{KL}(g_1 \mid g_2).$$

## J  ADDITIONAL EXPERIMENT DETAILS

We follow the data-generating process in Kallus et al. (2021) and Cui et al. (2020) to generate multi-dimensional variables $U, S, W, Q, A$ with $A \in \{0, 1\}$ as follows:

1. $S' \sim \mathcal{N}(0, 0.5I_{d_S})$, where $I_d$ is a $d$-dimension identity matrix.

2. $A|S' \sim \text{Ber}(p(S'))$ where

$$p(S') = \frac{1}{1 + \exp(0.125 - 0.125\mathbf{1}_d^\top S')},$$

   where $\mathbf{1}_d$ is all-one vector.

3. Draw $W', Q', U$ from

$$W', Q', U \mid A, S' \sim \mathcal{N}\left(\left[\begin{array}{c} \mu_0 + \mu_a A + \mu_s S' \\ \alpha_0 + \alpha_a A + \alpha_s S' \\ \kappa_0 + \kappa_a A + \kappa_s S' \end{array}\right], \left[\begin{array}{c} \sigma_w^2, \sigma_{wq}^2, \sigma_{wu}^2 \\ \sigma_{wq}^2, \sigma_q^2, \sigma_{qu}^2 \\ \sigma_{wu}^2, \sigma_{qu}^2, \sigma_u^2 \end{array}\right]\right).$$

   Here we set the parameters above as $\mu_0 = \alpha_0 = \kappa_0 = 0.2\mathbf{1}_d, \alpha_a = \kappa_a = \mu_s = \alpha_s = \kappa_s = \mathbb{I}_d, \sigma_q^2 = \sigma_u^2 = \sigma_w^2 = 0.1\left(\mathbb{I}_d + \mathbf{1}_d\mathbf{1}_d^\top\right), \sigma_{wu}^2 = \sigma_{zu}^2 = 0.1\mathbf{1}_d\mathbf{1}_d^\top$. Finally, we choose $\sigma_{wq}^2$ and $\mu_a$ to ensure that $W' \perp (A', Q') \mid U, S'$, which is a prerequisite of proximal causal inference (Kallus et al., 2021, Condition 4 in Assumption 1). To achieve this, note that

$$\mathbb{E}\left[W' \mid U, S', A, Q'\right] = \mu_0 + \mu_a A + \mu_s S' + \Sigma_{w(q,u)}\Sigma_{q,u}^{-1}\left[\begin{array}{c} Q' - \alpha_0 - \alpha_a A - \alpha_s S' \\ U - \kappa_0 - \kappa_a A - \kappa_s S' \end{array}\right] \tag{19}$$

   where

$$\Sigma_{w(q,u)} = \left(\sigma_{wq}^2, \sigma_{wu}^2\right), \quad \Sigma_{q,u} = \left[\begin{array}{c} \sigma_q^2, \sigma_{qu}^2 \\ \sigma_{qu}^2, \sigma_u^2 \end{array}\right].$$

   We simply select $\sigma_{wq}^2$ and $\mu_a$ so that Equation equation 19 does not depend on $A$ and $Q'$.

4. Draw $Y$ from

$$Y \mid X', U, W' \sim \mathcal{N}\left(A + \mathbf{1}_d^\top S' + \mathbf{1}_d^\top U + \mathbf{1}_d^\top W', 1\right).$$

5. Set $W' = W'_{[0:d_W]}$. Observe $S = g(S')$, $Q = g(Q')$, $W = g(W')$, where $g(\cdot)$ is a reversible function that operates component-wise on each variable.

Our data-generating process is described in Figure 1.

**Additional Numerical Results**

### J.1 HYPERPARAMETER SETTINGS.

For RDIV, we use Adam as the optimizer for both density estimation and Tikhonov regression, with a default learning rate of $10^{-4}$, a batch size of $50$, and a training epoch of $300$. All results are run on a 32GB CPU. We will show how to choose these hyperparameters with our model selection procedure (Algorithm 2) in Section J.2. For all baselines except for AGMM, we adapt the hyperparameters in their original codebase. For AGMM, we tune the learning rate for the learner and adversary for every $g(\cdot)$ independently. We follow Singh et al. (2019) to use Gaussian RKHS for function approximation and their method for tuning the regularization parameter. When $n = 500$, the learning rate of the learner and adversary in AGMM are manually set to $10^{-4}$ for LogSigmoid, Piecewise, and Sigmoid, and $10^{-3}$ for Id, Poly, and CubicRoot. When $n = 1000$, the learning rate of the learner and adversary in AGMM are manually set to $10^{-4}$ for Piecewise and Sigmoid, and $10^{-3}$ for LogSigmoid, Piecewise, and CubicRoot. The training parameter of DFIV is adopted from Xu et al. (2021). Note that tuning DFIV is highly intractable in practice, as their method is essentially a bilevel optimization, which is known to be hard to solve (Hong et al., 2023).

### J.2 MODEL SELECTION

While it is seen that Kernel IV is comparable to RDIV in some scenarios such as in Table 4 and 5, in this section, we will show that our RDIV equipped with a model selection procedure can generally outperform KernelIV. We report our results in model selection for the second stage by implementing Best-ERM in Algorithm 2 and demonstrate how it improves our results. Specifically,

our models $h_1, \ldots, h_M$ are trained by different hyperparameters. First, we employ model selection for the density function by Best ERM. Then with the trained density function in the first stage, we further apply Best ERM to the models in the second stage. In the model selection experiments, we fix the dimension of our dataset to be $d_S = d_Q = 20$, $d_W = 10$. We compute the mean and confidence interval with 10 independent trials. We set the candidate training parameters as follows: the number of epochs $\in \{300, 400\}$, the batch size for the 1st stage $\in \{30, 50\}$ and the batch size for the 2nd stage $\in \{50, 60, 100\}$, the learning rate $\in \{10^{-4}, 10^{-3}\}$, the number of mixture components $\in \{40, 50, 60\}$. As shown in Table 7, when RDIV is equipped with model selection techniques, our method outperforms KernelIV in all but one case when the dataset size is 500, and outperforms KernelIV in 3 out of 6 settings when the dataset size is 1000. Our approach demonstrates its effectiveness by outperforming previous benchmarks across a diverse set of Data Generating Processes (DGP). This achievement is attributed to both the ease of optimization of RDIV and its theoretically sound integration with model selection procedures.

