# OpenReview forum: "Regularized DeepIV with Model Selection"
_ICLR.cc/2025/Conference — Submitted to ICLR 2025_

### Official Review · Reviewer_P9wu · 2024-10-26

**Soundness:** 2
**Presentation:** 2
**Contribution:** 2
**Rating:** 5
**Confidence:** 4

**Summary:**

This manuscript is basically a technical paper, discussed about two-stage non-parametric IV and model-selection in the second stage when equipped with an additional $L_2$ regularization.

**Strengths:**

* The authors discussed lots of the aspects for non-parametric clearly.

**Weaknesses:**

* I feel the problem studied in this paper is with limited novelty. The transformation between one-stage and two-stage algorithms and analysis is in general only a technical problem and have been discussed in different places like DualIV, and the $L_2$ regularization itself makes the model-selection easier (with the strongly convexity).

**Questions:**

* To convert the MLE guarantee into an $L_2$ guarantee, the authors assumed a minimum density on the conditional density. What’s the benefits/drawbacks compared with the conditional mean embedding based methods (although it also requires some assumptions like HS operators).

---

> ### Author Response · Authors · 2024-11-20
> **Thank you for your review!**
>
> >**Q1**: The transformation between one-stage and two-stage algorithms and analysis is in general only a technical problem and has been discussed in different places like DualIV, and the L2 regularization itself makes the model selection easier (with the strong convexity).
>
> **A1**: Thank you for your comment! We agree that in one-stage minimax optimization problems, such as DualIV, the population loss shares the same estimator as two-stage methods, as the former is a dual transformation of the latter. However, we respectfully disagree with the assertion that “the transformation between one-stage and two-stage algorithms is redundant for general function approximation”.
>
> Converting IV estimations to its dual form (e.g., DualIV [1], AGMM [2]) introduces a minimax optimization framework, which is **theoretically intractable in computation and empirically challenging** for important function classes such as random forests or neural networks. Note that DualIV only studies kernel function approximation. Our contribution is that we rigorously prove that RDIV achieves comparable statistical convergence rates to minimax methods while being easily **applicable to general function classes** since it only requires single-level optimization, which has been widely studied and well-known to be tractable for function classes such as neural networks, thereby enjoying both theoretical rigor and practical feasibility.
>
> We also confirm with the reviewer that $L_2$ regularization plays a crucial role in our analysis for both the RDIV method and its model selection procedure. However, we would like to disagree that such regularization makes the model selection trivial. **Previous literature ([2][3]) studying minimax estimators for IV regression also incorporates $L_2$ regularization, but no guarantee for model selection has been established.** This is because it is hard to clarify the error caused by the model misspecification of the dual function in the minimax optimization. On the other hand, RDIV only relies on single-level optimization and thus enjoys a more tractable model selection. To our knowledge, RDIV is the first method that enjoys a provable guaranteed model selection procedure.
>
> >**Q2**: To convert the MLE guarantee into an L2 guarantee, the authors assumed a minimum density on the conditional density. What are the benefits/drawbacks compared with the conditional mean embedding-based methods (although it also requires some assumptions like HS operators)?
>
> **A2**:
> Thank you for bringing this to our attention! In the following, we will explain the motivation behind this technical assumption, provide a detailed comparison with mean embedding-based methods in [6], and show that our method is more general.
>
> (a) We confirm that we do assume a lower bound for the density when establishing the convergence rate of MLE in Assumption 4. However, such an assumption does not directly affect our analysis of the bias-variance tradeoff in our final result, see Lemma 1 and the proof of Theorem 5. We humbly point out that such an assumption is conventional when analyzing the convergence rate of MLE, and refer the reviewer to [4, Chpt.14.4.1] for details. This assumption can be easily relaxed by applying $\chi^2$-MLE instead of MLE as the density estimator, which we also discussed in Appendix C. A more recent line of work [5] provides a similar bound that does not require a lower bound of the density and can be easily incorporated into our proof.
>
> (b) The conditional mean embedding learned by the first stage in [6] is defined by the conditional expectation of the kernel function of $X$ under a given $Z$, thus is only defined for a kernel function class. When it comes to general function classes such as neural networks, such a function is usually undefined or intractable when the underlying Hilbert space is unknown. Moreover, [6] assumes realizability for the mean embedding, thus providing no results on how to perform model selection for the kernel classes when model misspecification exists. Compared with the mean embedding, RDIV (i) allows convergence guarantee for **general function classes beyond kernel classes**, (ii) uses MLE to estimate the conditional density in the first stage, thus **allowing establishing model selection results**. Therefore, we believe that the MLE method RDIV is more general than the mean-embedding-based method adapted by [6].
>
> [1] Dual Instrumental Variable Regression, Krikamol Muandet, Arash Mehrjou, Si Kai Lee, Anant Raj
>
> [2] Minimax Estimation of Conditional Moment Models, Nishanth Dikkala and Greg Lewis and Lester Mackey and Vasilis Syrgkanis. NeurIPS 2020
>
> [3] Source Condition Double Robust Inference on Functionals of Inverse Problems, Bennett et al.
>
> [4] High-dimensional statistics: A non-asymptotic viewpoint, Martin J. Wainwright.
>
> [5] Minimax Rates for Conditional Density Estimation via Empirical Entropy, Bilodeau et al.
>
> [6] Kernel Instrumental Variable Regression, Singh et al.

---

### Official Review · Reviewer_HEwd · 2024-10-31

**Soundness:** 3
**Presentation:** 4
**Contribution:** 3
**Rating:** 6
**Confidence:** 3

**Summary:**

The paper studies the nonparametric instrumental variable regression with Tikhonov regularization (RDIV), and proves that RDIV allows model selection procedures and matches the SOTA convergence rate. I agree with the author's claim that this work is the first attempt to provide rigorous guarantees for DeepIV. With Tikhonov regularization, the model selection procedure achieves the oracle rate and iterative RDIV matches the SOTA rate.

**Strengths:**

The paper is well-written, and the results are motivated well. I didn't go through the proofs, but the explanations after each result are insightful, and ease the reading.  The theoretical contribution is solid.

**Weaknesses:**

The numerical experiments are only based on simulated data. It would be better to have some results from real data to demonstrate the strength of the proposal.

**Questions:**

How is Tikhonov regularization related to a function space parametrized by the neural network? It seems not straightforward to relate it to weight decay.

Is there a computational gain when minimax optimization is no longer needed?

---

> ### Author Response · Authors · 2024-11-20
> **Thank you for your review!**
>
> >**Q1**:How is Tikhonov regularization related to a function space parametrized by the neural network? It seems not straightforward to relate it to weight decay.
>
> **A1**: Thanks for bringing this to our attention! We confirm that Tikhonov regularization is different from weight decay, the latter being an $L_2$ regularization on the neural network parameters instead of the function class. In our experiments, we calculated the empirical mean of $h_\theta(X)^2$ by forward propagation and incorporated this into our loss function when training the model.
>
> >**Q2**: Is there a computational gain when minimax optimization is no longer needed?
>
> **A2**: We confirm that there is an obvious computational gain when compared with classical minimax methods such as AGMM because our method is easy to tune, allows model selection, and is thus more tractable. We humbly refer the reviewer to our Section 9 and Appendix J for more details.

---

### Official Review · Reviewer_q5qB · 2024-11-04

**Soundness:** 3
**Presentation:** 3
**Contribution:** 3
**Rating:** 5
**Confidence:** 4

**Summary:**

This paper addresses the problem of nonparametric instrumental variable (IV) regression, a framework with wide applications across fields such as causal inference, handling missing data, and reinforcement learning. The objective in IV regression is to solve the conditional moment equation, 𝐸 [ 𝑌 − ℎ ( 𝑋 ) ∣ 𝑍 ] = 0, where 𝑍  serves as the instrument. The authors introduce RDIV, a regularized variant of the DeepIV method, marking the first work to provide rigorous theoretical guarantees for DeepIV. RDIV enhances generalization by incorporating Tikhonov regularization. Methodologically, RDIV follows a two-stage approach. The first stage involves learning the conditional distribution of covariates, while the second stage refines the estimator by minimizing a Tikhonov-regularized loss function.

**Strengths:**

RDIV offers several key advantages over existing methods. It addresses three significant limitations of prior literature: it eliminates the need for unique IV regression identification, avoids reliance on the often unstable minimax computation oracle, and supports model selection procedures.

**Weaknesses:**

It is unclear how the method compares for example to recently developed methods (see arxiv:2405.19463; to appear at NeurIPS 2024) that completely avoids minimax formulations, as well as avoiding the need for two-stage procedures.

**Questions:**

please see above

---

> ### Author Response · Authors · 2024-11-20
> **Thank you for your review!**
>
> >**Q1**: It is unclear how the method compares for example to recently developed methods (see arxiv:2405.19463; to appear at NeurIPS 2024) that completely avoid minimax formulations, as well as avoid the need for two-stage procedures.
>
> **A1**: Thank you for bringing the paper to our attention! We would like to highlight a few key differences between our works and [1]:
> Our work studies the case where both the model class $\mathcal{H}$ and conditional density function class $\mathcal{G}$ are **general function classes**, which includes random forest, neural network, and parametric models as special examples, and establishes **a provable statistical convergence rate**, while [1] studies parametric function classes, including linear function class and general linear function class with a known link function $g$, and derived convergence rate based on their specific optimization scheme. However, their techniques are not directly transferable to general function approximation. Moreover, no guarantee or procedure for model selection for function $g$ is discussed in [1].
> When facing a nonlinear link function $g$ in TOSG-IVaR (Alg.1 of [1]), [1] assumes that we have access to a two-sample oracle that outputs two samples $X$ and $X$′ that are independent conditioned on the same instrument $Z$. Meanwhile, our results for general nonlinear function classes do not need such an assumption, which is often impractical.
> When restrained our study to a parametric function class as in [1], our result in Theorem 5 provides a guarantee of $$||\hat{h} - h_0||_2 \leq \tilde{O}(\sqrt{1/n}),$$ since $\beta = +\infty$ and $\delta_n = O(\sqrt{1/n})$. This aligns with Theorem 2 in [1].
>
> We will upload an updated version of our work and summarize the comparison between [1] and our work in the Related Work section.
>
> [1] Stochastic Optimization Algorithms for Instrumental Variable Regression with Streaming Data, Xuxing Chen, Abhishek Roy, Yifan Hu, Krishnakumar Balasubramanian

---

### Official Review · Reviewer_MFbt · 2024-11-05

**Soundness:** 3
**Presentation:** 2
**Contribution:** 3
**Rating:** 5
**Confidence:** 3

**Summary:**

This paper studies a two stage procedure for regression in the scenario where the errors are not conditionally independent. They first learn a conditional density to make use of instrumental variables and consequently solve a square loss erm problem weighted by the learned conditional density. They show that this procedure attains mostly standard nonparametric rates.

**Strengths:**

* With the (unfortunate) exception of the introduction, I found the paper mostly well-written and clear.

* The paper studies an interesting problem, proposes a natural solution, and proceeds to analyze said solution. While I am not familiar with the immediately preceding related work (in IV), this seems clean to me.

**Weaknesses:**

* The organization of the paper is hard to follow and the introduction is way too terse. As someone well-versed in nonparametric statistics but not necessarily IV methods, I had to skip ahead to section 4 to really understand what was going on.  Stating that you are trying to solve some fixed point equation in the introduction is not conducive to most people's understanding of the problem you are solving.

* My overall feeling is that the result is somewhat incremental. To my understanding the main difficulty lies making standard guarantees for MLE in Hellinger^2 compatible with the square loss. I could not entirely follow why this is so challenging and would encourage the authors to further explain why this is the case (for instance, in the very last paragraph of section 1, you mention this difficulty but do not really expand on it, nor do you reference the lemmata which might be useful for understanding this difficulty).

**Questions:**

* is it really fair to say that your algorithm is more computationally tractable when it is based on MLE?

---

> ### Author Response · Authors · 2024-11-20
> **Thank you for your review!**
>
> >**Q1**: The result is somewhat incremental. To my understanding, the main difficulty lies in making standard guarantees for MLE in Hellinger^2 compatible with the square loss. I could not entirely understand why this is so challenging, and I would encourage the authors to explain why this is the case further.
>
> **A1**: Thanks for bringing this to our attention! In the following, we will first elaborate on the technical challenges we tackled in our proof, and then we will highlight the contribution of our result to IV regression.
>
> (1) Our technique for bounding the $L_1$ norm instead of the $L_2$ norm of $(\mathcal{T} - \hat{\mathcal{T}})(h - \hat{h})$ is essential for the fast rate result. The standard guarantee of MLE ensures that $H(\hat{p}, p)^2 \leq \delta_n^2$, directly applying which to the $L_2$-norm will result in a bound of
> $$||(\mathcal{T} - \hat{\mathcal{T}})(h - \hat{h})||_2 \leq \delta_n^{1/2} ||h - \hat{h}||_2, $$
>
> or
>
> $$||(\mathcal{T} - \hat{\mathcal{T}})(h - \hat{h})||_2\leq\delta_n||h - \hat{h}||\infty, $$
>
>
>  plugging both bounds back to our proof for Theorem 5 on page 19 will bring in a **slower rate** in the bias-variance tradeoff, or further assumptions on the relative smoothness between $||\cdot\||_\infty$ and $||\cdot||_2$ (e.g. $\gamma$-smoothness, see Remark 2 in [1] ).  On the contrary, **we derived a tighter bound on the $L_1$ norm of the error and incorporated this with our strong convexity analysis of the Tikhonov estimator**. This is non-trivial, as the standard analysis in nonparametric statistics usually only provides bounds in $L_2$-norm.
>
> (2) We would like to iterate our main contribution from two perspectives:
> As introduced in our introduction, our work establishes **the first theoretical guarantee for the popular DeepIV type method [2]**, which has been lacking for years. We also point to a crucial modification that is required for that method to be made rigorously efficient (regularization) from both theoretical and empirical studies. Although simple, such a modification allows RDIV to enjoy a provably convergence guarantee that almost matches the SOTA convergence rate, meanwhile does not introduce an intractable minimax optimization.
> Thanks to its theoretical guarantee, we establish a **theoretically grounded model selection procedure** for RDIV in Sec. 7, namely the Best-ERM and the Convex-ERM methods. Those procedures enable us to select the function class or the training hyperparameter for RDIV, and are widely discussed in statistics and machine learning [5][6]. Notably, existing methods such as the original DeepIV, DFIV [3], or AGMM [4] **do not have any guarantee** for model selection. Moreover, we empirically show its effectiveness with proximal causal inference as an example and summarize our result in Section 9 and Appendix J.
>
> > **Q2**: Is it really fair to say that your algorithm is more computationally tractable when it is based on MLE?
>
> **A2**: Thanks for your helpful comment! We believe that the application of MLE does not hurt the tractability of our method. In our study, we have already conducted experiments on low-dimensional and median-dimensional data, in which MLE is naturally tractable by function approximation such as Gaussian mixture models. For more specific high-dimensional data such as images, since our method only requires sampling from $P(x|z)$,  Step 1 could be easily conducted by Flow-based models [7] or diffusion models [8].
>
>
> [1] Source Condition Double Robust Inference on Functionals of Inverse Problems, Bennett et al.
>
> [2] Deep IV: A Flexible Approach for Counterfactual Prediction, Jason Hartford, Greg Lewis, Kevin Leyton-Brown, Matt Taddy. ICML 2017
>
> [3] Learning Deep Features in Instrumental Variable Regression, Liyuan Xu and Yutian Chen and Siddarth Srinivasan and Nando de Freitas and Arnaud Doucet and Arthur Gretton. ICLR 2021
>
> [4] Minimax Estimation of Conditional Moment Models, Nishanth Dikkala and Greg Lewis and Lester Mackey and Vasilis Syrgkanis. NeurIPS 2020
>
> [5] Peter L Bartlett, Stéphane Boucheron, and Gábor Lugosi. Model selection and error estimation. 353 Machine Learning
>
> [6] Guillaume Lecué and Philippe Rigollet. Optimal learning with q-aggregation. Annals of Statistics 2014.
>
> [7] Danilo Jimenez Rezende, and Shakir Mohamed. “Variational inference with normalizing flows.” ICML 2015.
>
> [8] Denoising Diffusion Probabilistic Models, Jonathan Ho, Ajay Jain, Pieter Abbeel.

---

### Author Response · Authors · 2024-11-27
**Thank you for your review**

Dear Reviewers,

We sincerely appreciate the time and effort you have dedicated to reviewing our paper. During the initial phase of the rebuttal process, we carefully addressed your concerns and provided an updated version of the manuscript. We would like to reiterate our response in the following:

>Q1. The technical novelty of this paper, and the contribution it made to IV regression

**A1**. (1) Our technique for bounding the $L_1$ norm instead of the $L_2$ norm of $(\mathcal{T} - \hat{\mathcal{T}})(h - \hat{h})$ is essential for the **fast rate result**. Particularly, directly applying the Hellinger distance to $L_2$-norm results in a bound of $||(\mathcal{T} - \hat{\mathcal{T}})(h - \hat{h})||_2 \leq \delta_n^{1/2} \cdot ||h - \hat{h}||_2$, which further results in a slower convergence rate of $|| h_0 - h^*||_2 \leq O(\delta_n^{\frac{\min(\beta,2)}{\min(\beta,2) + 2}})$. Our approach in Lemma 1 and Theorem 5 manages to bound $\hat{\mathcal{T}})(h - \hat{h})||_1 \leq O(\delta_n \cdot || \hat{h} - h ||_2)$, and carefully applying this result to the second stage guarantee results in a rate of $|| h_0 - h^*||_2 \leq O(\delta_n^{\frac{2\min(\beta,2)}{\min(\beta,2) + 2}})$, which is way much faster than the trivial result obtained with $L_2$ norm and comparable to the existing state of the art.

(2) Our work establishes the **first theoretical guarantee for the popular DeepIV type method** [2], which has been lacking for years, and establishes a statistical convergence rate that **almost matches the SOTA convergence rate** in existing minimax approaches while remaining **computationally tractable**. We also point to a crucial modification that is required for that method to be made rigorously efficient (regularization) from both theoretical and empirical studies. Moreover, we establish a **theoretically grounded model selection** procedure for RDIV in Sec. 7, namely the Best-ERM and the Convex-ERM methods. Although widely discussed in machine learning literature, a provably efficient model selection guarantee has been lacking in existing works such as DeepIV, AGMM [4], and DFIV.

>Q2. Comparison with TOSG-IVaR [2] and DualIV [3]

**A2**. In our paper, we studied RDIV under **general function approximation**, and how to perform the model selection procedure, with both guarantees from theoretical and empirical perspectives. In contrast, [2] only studies stochastic optimization for IV regression under **parametric function classes**, including linear and generalized linear function classes, which are less general compared to our setting. Based on such a setting, the optimization scheme proposed in [2] cannot be directly extended to nonparametric function classes. When our function class is restricted to parametric function classes, RDIV allows a convergence rate of $O(1/ \sqrt{n})$, identical to [2].

[3] studies minimax optimization within a **kernel function class**. However, their method (i) provides no convergence guarantee, (ii) suffers computational intractability when transferred to the general function class due to the absence of a closed-form solution for the dual problem, and (iii) does not provide a theoretically grounded model selection procedure. Notably, [3] is a special case of the AGMM method [4], the latter studies minimax methods under general function approximation, and is discussed in detail in our related work section.

>Q3. Computational efficiency for MLE

**A3**. We believe that the application of MLE does not hurt the efficiency of our method. In our study, we have already conducted experiments on low-dimensional and median-dimensional data, in which MLE is naturally tractable by function approximation such as Gaussian mixture models. For more specific high-dimensional data such as images, since our method only requires sampling from $P(x|z)$,  Step 1 could be easily conducted by Flow-based models [5] or diffusion models [6].

As we approach the midpoint of the rebuttal period, we are eager to hear your feedback on the revised paper. We are committed to addressing any remaining questions or concerns you may have and further improving the work based on your valuable insights.

Thank you again for your thoughtful and constructive feedback.

[1] Deep IV: A Flexible Approach for Counterfactual Prediction, Jason Hartford, Greg Lewis, Kevin Leyton-Brown, Matt Taddy. ICML 2017

[2] Stochastic Optimization Algorithms for Instrumental Variable Regression with Streaming Data, Xuxing Chen, Abhishek Roy, Yifan Hu, Krishnakumar Balasubramanian

[3] Dual Instrumental Variable Regression, Krikamol Muandet, Arash Mehrjou, Si Kai Lee, Anant Raj

[4] Minimax Estimation of Conditional Moment Models, Nishanth Dikkala and Greg Lewis and Lester Mackey and Vasilis Syrgkanis. NeurIPS 2020

[5] Danilo Jimenez Rezende, and Shakir Mohamed. “Variational inference with normalizing flows.” ICML 2015.

[6] Denoising Diffusion Probabilistic Models, Jonathan Ho, Ajay Jain, Pieter Abbeel.

---

### Meta-Review · Area_Chair_YW95 · 2024-12-21

**Metareview:**

This study represents a solid and good contribution. However, the novelty and usefulness of the approach have not been fully appreciated, at least by the reviewers. The paper's improvements in norm selection and error evaluation are not presented in a form that is easily accepted by the majority of the machine learning community. The lack of experiments with real and large data sets is also noted as a problem. Since this is solid and good research, it would be necessary to make it more appealing to the community.

**Additional Comments On Reviewer Discussion:**

MFbt commented on the limited contribution in addition to writing problems; P9wu also pointed out the lack of novelty; q5qB pointed out the need for comparison with previous studies; HEwd insisted on the need for experiments with real data; and MFbt suggested that the study should be expanded to include more data.

---

### Decision · Program_Chairs · 2025-01-22

Reject